# Sharing the effort of the European Green Deal among countries

Karl W. Steininger 1,2✉, Keith Williges 1,3✉, Lukas H. Meyer 3, Florian Maczek4 & Keywan Riahi 4

In implementing the European Green Deal to align with the Paris Agreement, the EU has raised its climate ambition and in 2022 is negotiating the distribution of increased mitigation effort among Member States. Such partitioning of targets among subsidiary entities is becoming a major challenge for implementation of climate policies around the globe. We contrast the 2021 European Commission proposal - an allocation based on a singular country attribute - with transparent and reproducible methods based on three ethical principles. We go beyond traditional effort-sharing literature and explore allocations representing an aggregated least regret compromise between different EU country perspectives on a fair allocation. While the 2021 proposal represents a nuanced compromise for many countries, for others a further redistribution could be considered equitable. Whereas we apply our approach within the setting of the EU negotiations, the framework can easily be adapted to inform debates worldwide on sharing mitigation effort among subsidiary entities.

[1] Wegener Center for Climate and Global Change, University of Graz, A-8010 Graz, Austria. [2] Department of Economics, University of Graz, A-8010 Graz, Austria. [3] Department of Philosophy, University of Graz, A-8010 Graz, Austria. [4] Energy, Climate and Environment Program, International Institute for Applied Systems Analysis, A-2361 Laxenburg, Austria. ✉email: karl.steininger@uni-graz.at; keith.williges@uni-graz.at

Most climate goals are framed in terms of very broad scales, such as the implicit global emissions reduction target of the Paris Agreement[1,2], but these overall targets need to be divided amongst lower levels such as states, provinces or sectors of societies[3–5]. In political processes, quite often this effort sharing of a common target is hampered by diverging viewpoints and perspectives on how to determine a fair share.

Considering the case of the European Union, while all large installations are regulated within the EU Emission Trading System (ETS) under an EU-wide cap, the majority of greenhouse gas emissions is tackled under the EU 'effort-sharing' approach, which consists of individual binding emission reduction targets for Member States. Basically, the EU has implemented the regulatory approach of the Kyoto Protocol—which included individual emission reduction commitments as well as various emission trading options (flexibility instruments)—among its Member States[6]. Initially, the so-called Triptych sectoral approach[7] differentiated three economic segments: (i) the power sector, (ii) the energy-intensive industry exposed to international trade and (iii) the remaining domestically-oriented sectors, with emission reductions for each calculated by application of different rules (see Supplementary Information section 1 for details). For the domestic sectors, a per capita emission allowance approach was used. Building upon this experience (considered to have successfully contributed to political acceptance[8]) the regulatory effort-sharing framework to cover emissions outside the EU ETS was further developed by the Effort Sharing Decision and the Effort Sharing Regulation, adopted in 2009 and 2018, respectively. In terms of the criterion used to set differential national targets, both mainly relied on economic performance, i.e. capability as measured in terms of GDP/capita. In 2021 an overall target of 55% reduction by 2030 compared to 1990 was established[9], but the distribution of that effort among Member States is undecided and under negotiation in 2022.

In this work, we present an approach that can contribute to a transparent decision process when partitioning an overall emissions target among subsidiary entities by aligning disparate views and defining an allocation space where different parties could agree to an equitable compromise. It builds upon the categorization of equity principles used in the effort-sharing literature[10–13]. For negotiation processes heavily influenced by a range of factors, such as historical path dependencies[14], side deals, power struggles, different cultures, or domestic politics, such transparency in terms of equity may be a significant factor contributing to success, given the increasing weight of the equity dimension in such negotiations[15]. We assess the implications of the proposed approach—in terms of the resulting emissions budget allocation and corresponding reduction targets—in the context of the European Union (EU) negotiations in 2022 to raise the 2030 effort-sharing target[9]. We first explore the implications of different effort-sharing mechanisms and interpretations of equity or justice on a singular, country level, finding that the resulting ranges of possible emissions reduction burden can vary widely. In such a case, if each country were to favor a different equity interpretation, the overall target of the EU in 2030 would most likely remain unmet[16,17]. However, our proposed framework allows for systematically combining different interpretations of equity or justice on the basis of three major principles—capability, equality, and responsibility—and allows us to explore situations where country targets are the result of a weighted combination of interpretations of those principles, where member state reductions add up to the overall EU reduction target. In doing so we distinguish interpretations of equity or justice that reflect considerations of equality, capability and responsibility. By identifying weighted combinations of interpretations of these principles that minimize the changes in emission reduction effort required by countries compared to their (i) upper bounds of equity-compatible emissions or, given the practical importance of negotiation history, (ii) previous commitments in the earlier less ambitious effort-sharing agreement, we identify a possible space for decision-making which combines multiple equity interpretations. We find that the possible combinations of equity interpretations, and the strength at which they are applied, result in a wide range of space for decision-making which comprises a richer set of ethical considerations than the 2021 proposed EU approach. We conclude with a discussion of the described framework and its applicability in specific negotiation and policy-making processes, also in other international contexts.

## Results

**Equity principles**. In analyzing how the EU emissions budget 2020–2030 can be allocated among the EU 27 we distinguish three principles, namely—following the IPCC's broad classification[18]—capability, equality, and responsibility and different interpretations of each of these principles. According to this understanding, these principles can be interpreted differently as they can reflect different ethical considerations. Different interpretations of these principles and combinations of them amount to different interpretations of what allocation can be considered equitable or just. Our approach builds upon the state of the art—in particular as reflected in the IPCC, which is focused on informing policy makers—and aims to assess the implications of imposing equity or justice principles in negotiations on the allocation of emission reductions within the EU.

According to the capability principle the greater an agent's ability to pay for the solution of a problem (in this case, reaching the 2030 EU reduction goal) the greater the proportion of costs that the agent should be expected to pay[19–24]. This principle, often dubbed the ability to pay principle, reflects an egalitarian understanding of justice[25]. The principle relies on the idea of positive duties of the most and more advantaged to help those less advantaged and worst off. The principle in itself does not take into account who can be held responsible for high current emission levels and who has so far been more or less benefited by emission-generating activities. To operationalize the principle, the indicator most often employed in the literature is GDP per capita, serving as proxy for differing abilities to pay. The magnitude at which differences in this type of ability translate to differences in required emission reduction is implemented here via two different approaches (*C1-EU-capability* and *C2-GDP/cap*; for further detail, see Table 1 and the methodology section). Both approaches are similar in their translation of differences in GDP per capita levels into emission reduction needs according to the capability principle, but of note is that C1 is designed to mirror as closely as possible the 2021 EU proposal[9].

While interpretations focusing on macroeconomic indicators do address the ability to pay in a very literal sense, the ability of an actor can be argued to extend beyond GDP. Issues of governance, and the ability of an actor to effect changes, rely also on institutional effectiveness, human capital, bureaucratic quality, and other aspects not taken explicitly represented by GDP. To incorporate these, indicators of government effectiveness (such as ref. [26]) can be used to incorporate differences in perceived quality of public and civil services or quality of policy formulation to emission reduction needs (*C3-Governance*, see Table 1 and SI section 2 for details).

Alternatively, the capability to reduce emissions in the future could be reflected in recent achievements in building up renewable energy capacity. The expansion of renewables could indicate increased ability to reduce emissions relative to other

**Table 1 Alternative interpretations of the three equity principles (responsibility, capability, and equality) as applied in this work, in line with ref. [10] Table 6.5.**

| Interpretation | Relevance and operationalization |
|---|---|
| *Capability* | |
| C1- EU implementation (*EU-capability*) | Corresponds to the 2021 EU policy proposal that puts a cap on the relevance of countries' per capita GDP differences for the required emission reductions and is derived empirically from the 2021 proposal. The distribution is estimated in a regression based on GDP per capita from 2015–2018 and the previous (2018) ESR distribution (which builds on GDP/capita). |
| C2- GDP per capita (*GDP/cap*) | Weighs the relevance of all per capita GDP differences equally in specifying countries' required emission reductions, increasing or reducing emission allocations based on the deviation from the EU-average GDP per capita. Countries with a higher GDP per capita (in 2019) are allocated a smaller emission budget, i.e., a stricter emission reduction target. For each percent above or below average, reductions are increased (reduced) by an equivalent share. |
| C3- Government effectiveness (*Governance*) | Takes into account additional factors which contribute to the ability of countries to reduce emissions using a governance indicator (government effectiveness). Similarly to C2, countries with higher indicator values are allocated a smaller emission budget. The indicator takes into account a number of considerations, from institutional effectiveness to infrastructure, human capital, and policy efficacy. |
| C4- Renewable growth capacity (*RES-cap*) | Reflects the ability of countries to reduce emissions via development of renewable energy sources, similar to C3. As countries that have recently (since 2005) more strongly developed their RES capacity are more likely to have the capital, institutional framework, and first-mover advantages to enable further growth in the future, we allocate greater emissions reduction burdens to those countries, and comparatively less reductions to countries with less capacity. |
| *Equality* | |
| E1- Basic needs | Separates allocation of emission budgets into two stages. States are allocated emissions required to meet the basic needs energy demands of the fraction of the population at risk of poverty to reflect the satisfiers of meeting needs as people typically require, as meeting such needs cannot be directly measured. Second, the remainder of the budget is distributed in an equal-per-capita manner, so that all states are assigned at least enough emissions to reach the basic needs threshold, and then move beyond them. |
| E2- ES-sector EPC convergence (*ES-EPC*) | Reflects a convergence to equal-per capita emissions by 2030 (beginning at today's unequal levels of emissions), based on country emissions in effort-sharing (ES) sectors (i.e., emissions not covered under the Emission Trading System). |
| E3-Full-EPC convergence (*Full-EPC*) | Reflects convergence to equal-per capita emissions by 2030 (beginning at today's unequal level of emissions), based on all sectors' emissions (sectors in and outside the Emission Trading System). |
| *Responsibility* | |
| R1- Historical emissions from 1995 (*Hist-emi*) | Reflects emissions generation since 1995, when countries were liable to know the impacts of GHG emissions on climate and had the ability to abate. The point of time at which the remaining budget is allocated on an equal-per capita basis is shifted back to 1995. The EU GHG budget for 2020-2030 is extended by EU past emissions 1995-2019. Per capita budgets are allocated to countries as though budgeting began in 1995. The budget already used up by each country in 1995-2019 is subtracted to give the remaining national budgets for 2020-2030. |
| R2- Inherited benefits of emissions (*Benefits*) | Incorporates the benefits a country has obtained due to emissions prior to 1995, interpreted as the emissions embodied in national capital stock. Using capital stock estimates, GHG budgets are scaled similarly to R1 above, but based on pre-1995 emissions embodied in each country's capital stock in 1995. |
| R3- C-budget | The total emissions budget for the ES sector (calculated by a fictitious linear path from 2020 to 2030) is split among states according to population without any convergence period, thus eliminating any aspect of grandfathering. |
| R4 - Expansion of renewables (*RES expansion*) | Reflects the differing change in renewable share from 2005 to 2019 of countries compared to the EU average. Countries with a higher relative change are allocated a larger emission budget, i.e., a more relaxed reduction target. Similar to C2, countries receive more (less) emissions at an equal rate as their increase (decrease) in RES share relative to the EU average. |
| R5- Cumulative emissions per capita (*Budget/cap*) | Proposes an alternative method to address historic emissions as compared to R1, by scaling future emission allowances based on differences in historical cumulative emissions per capita. Countries with a higher than EU-average cumulative historic emissions per capita from 1995 to 2019 are assigned higher reduction targets in 2030, and vice versa. |

*italics* indicate the shorthand notation for interpretations used throughout text and figures. For full details, see "Methods".

countries, whether due to circumstances such as advantageous natural resources or technological know-how resulting from early adoption and the consequent ability to efficiently expand the use of renewable energy sources (RES) in the future (first-mover advantage). In that sense, RES expansion could be interpreted as an indicator of capability; when countries have succeeded in improving emission efficiency by means of implementing renewables, they could be considered more capable of further reducing emissions and, thus, should be allocated a smaller share

of the remaining budget. We thus include an interpretation placing greater emission reductions on countries most likely able to meet such demands, as represented by recent development of RES (*C4-RES-cap*).

According to the equality principle, everyone should be able to enjoy a level of wellbeing above the level required to secure basic needs[27–30]. Allocating the burdens of reaching the 2030 reduction goal should be compatible with countries securing the sufficiency level of wellbeing of all residents. Reaching the poverty line serves

as proxy for reaching this critical level of wellbeing. So understood, the equality principle reflects the sufficiency principle[31–34] (abbreviated *E1-basic-needs*). As an alternative to the sufficientarian interpretation, the equality principle can be specified as the goal of all countries converging on the same equal-per capita emission level in 2030 (defined as *E2-ES-EPC* or *E3-Full-EPC*, indicating convergence of effort-sharing (ES) or all (full) emissions). Then, from 2030 onwards the EU 27 will pursue the 2050 reduction of net-zero without grandfathering[35,36], that is, without prolonging the inequality of the status-quo levels of emissions into the transformation period.

According to the responsibility principle, states should be responsible for their own emissions since they have been liable to know about the limited capacity of the atmosphere to absorb greenhouse gases, their countries' share of the use of this limited resource and that all plausible understandings of sharing the remaining carbon budget require drastic reductions of emissions of most countries, including all of the EU 27. The latest plausible date for attributing such liability seems to be 1995, the date of publication of the second assessment report of the IPCC, identifying a discernible human influence on the climate beyond the findings of the first report of 1990, it's likely consequences, and, by doing so, suggesting that measures are required to hinder 'dangerous climate change'[37–40]. One could cite other dates for good reasons, e.g. the 1992 ratification of the UNFCCC at the Earth Summit or the late 1980s establishment of the IPCC, which we assess in more detail in the Conclusions section and in the Supplementary Information, section 3. By moving the year of accounting back to 1995 the actual emissions caused since then are attributed to the emitting countries. As interpretations to address historical considerations can take different forms, we utilize two approaches, indicated as *R1-hist-emi* and *R5-cumulative-emi/cap*, with their distinctions discussed in Table 1 and the methodology section.

The responsibility principle can also be specified in terms of taking into account the unequal benefits countries have received from the consequences of pre-1995 emission-generating activities[41]. Here the aim is to fairly distribute these benefits among currently living and future people[42] (*R2-Benefits* interpretation). The carbon emissions embodied in the countries' capital stock in 1995 serves as proxy for inherited benefits.

The third interpretation rejects the significance of the historically developed de facto unequal levels of per capita emissions among the EU 27 in 2020 and, instead, for the period of 2020–2030 relies on an equal-per-capita allocation of the overall EU-27 carbon budget to country budgets[43–45] (*R3-C-budget* interpretation). If so the EU 27 will pursue the 2030 reduction goal without grandfathering.

The fourth interpretation of the responsibility principle rewards countries' past efforts in improving emissions efficiency (i.e. raising output per emissions) by means of implementing renewables (*R4-RES expansion*), taking an alternative aspect of such expansion into account compared to the capability interpretation. In this instance, the justification is due to the argument that when countries succeed in improving emissions efficiency they can realize a higher level of wellbeing with the same emission budget. Justice is concerned with both fair shares of wellbeing and absolute levels of wellbeing not only in terms of all reaching the critical level, but also above the sufficiency level. In its economic interpretation both relative and aggregative welfare are important concerns. Among other principles, the so-called priority view takes into account both distributive and aggregative concerns[46]. Other things being equal (here if the likelihood of reaching the 2030 reduction goal remains the same) it is better when the EU 27 reach the goal with an on average higher per capita level of welfare, enabled by an expansion of

renewables. Given the high interdependencies within the EU it seems likely that welfare gains realized in one country will benefit people in other countries of the EU as well. Countries can be understood to be responsible for such renewable expansion measures since they have become liable for their emission-generating activities. The proxy for success of such measures—and granting an increased budget as a consequence—is the increase in the share of renewables.

The three equity principles discussed above—operationalized via their various interpretations—can be utilized individually to allocate emissions budgets, but they can also be applied together in varying degrees of intensity. Figure 1 provides a conceptual overview of such an approach. Each equity principle forms one corner of a triangle and is represented by an interpretation. The results of the chosen interpretations can be combined in a weighted sum to arrive at an allocation incorporating all three equity considerations.

**Country reduction targets for single equity interpretations**. We first apply the interpretations discussed in the previous section in a single manner, determining individual country emission reduction targets as specified in the Methods section. Figure 2 illustrates the resulting country emissions reduction burden for the twelve interpretations presented. The figure makes readily apparent the 2021 approach of the EU to allocate 2030 budgets using a GDP per capita-based allocation with some adjustments, as shown by the consistent proximity of our capability considerations to the proposed levels, barring a few outlier countries (e.g., Ireland and Luxembourg, with per capita GDP levels two to three times higher than the EU average). Beyond this, we are able to arrange the countries into three distinct groups; the first where the (majority of) equity-based allocations suggest a less stringent target than that proposed by the European Commission (EC), a second where no clear trend exists (results are above and below the EC proposal), and a third where equity-based allocations lead to a more stringent target. For clarity, we have organized these countries according to this trend, in panels A, B, and C.

For the first group of countries (Fig. 2A) where the allocation results tend to require *lower* levels of emissions reduction than the EC's proposal, a number of factors are at play. The 2021 proposal of the Commission mainly adjusts for country differences in GDP/capita, and thus acknowledges mainly a capacity consideration. In this first group, countries are allocated more emissions (i.e., lower emission reduction burden) for the bulk of the interpretations of the responsibility and equality dimensions, given these Member States are: (a) countries with a higher share of the population in poverty (beyond a mere low *average* GDP/capita), such as Bulgaria or Romania; and (b) countries for which responsibility interpretations result in less future reduction, most often according to all of its possible interpretations, be it low historical emissions (e.g. Bulgaria, Spain, Italy or Portugal), substantial success in emission reduction (Sweden) or renewables expansion (Sweden, Cyprus), or (for most of these countries: and) comparatively low emission per capita starting levels granting least reduction when the remaining carbon budget is allocated equally to individuals across countries.

For the second group of countries (Fig. 2B) at least two equity dimensions would lead to either a stricter of less stringent reduction target, dependent on and differing across their respective interpretation. For example, an acknowledgment of GDP per capita differences without the 50% emission reduction level cap of the EU commission proposal would require a more stringent reduction from Germany, as would an acknowledgment of inherited benefits, while past success in renewables extension and consideration of historical emissions (more specifically,

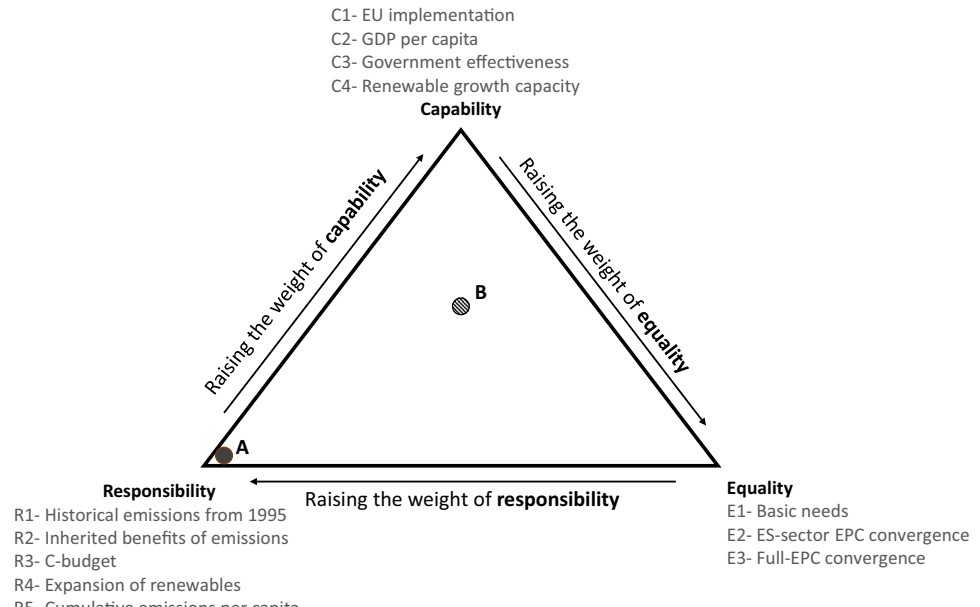

**Fig. 1 Conceptual overview of allocation approach.** Each distribution consists of three components, addressing each of the three IPCC equity considerations of (i) responsibility, (ii) capability, and (iii) equality. These equity considerations are interpreted via a set of interpretations, listed under each corner. The interpretations used in the main scenario are indicated as R1, E1, and C1; alternative interpretations are listed with subsequent numbers, e.g. R2, where the inherited benefits from pre-1995 emission-generating activities replaces historical emissions in the scenario calculations. As an example, Point A would indicate an allocation scenario where only responsibility is given weight. Point B represents an allocation where equal weight is given to all three equity considerations.

comparatively larger emission reduction already achieved) indicate the reverse, i.e., lower emission reduction target for this country. An equally divergent result, albeit exactly in the reverse direction for each of these interpretations holds for Hungary, Latvia, Croatia, and Poland. For all the countries in this group, the way the equity dimensions are specified matters for the direction they are impacted.

Finally, for the remaining group, consideration of the further equity dimensions will generally lead to stricter reduction targets than suggested by the EU proposal. For countries like Austria, Belgium, Denmark, Ireland, Luxembourg, and the Netherlands, and also Czechia and Slovenia, such consideration clearly increases the required reduction efforts. For some of these countries one or at most two interpretations work in the opposing direction: for Denmark and Finland a responsibility acknowledgment of past RES expansion would reduce, not increase, the reduction target, while for the Netherlands and Slovenia a capability interpretation of RES would do so.

Note, that in accordance with the history of global and EU emission negotiations all interpretations are based on production-based emission accounting. For historic responsibility (R1) quantitative results would differ, if based on the alternative consumption-based emission accounting, which allocates emissions of the full value chain to the country of final demand, irrespective of where the emissions physically have occurred during the production process. For the EU27 for all but four countries consumption-based emissions are higher—those four being Bulgaria, the Czech Republic, Denmark, and Poland[47,48]. Thus while a switch to consumption-based accounting would increase reduction commitments under R1 for the former countries, but decrease for the four latter ones, the classification of the three groups would not change.

Figure 2 thus indicates how strongly emission reduction obligations change for each EU Member State when following any given principle or their interpretations. It also makes clear that if countries were to choose the consideration and interpretation

that indicates the maximum equity-compatible emission level, i.e., the least emission reduction for them, the overall emissions reduction target for the EU for 2030 would be missed by a wide margin, in particular since interpretations can even be found for a majority of countries which would require zero reductions from 2005 levels.

**Implementing combinations of equity interpretations.** As laid out conceptually in Fig. 1, we can move beyond the use of a single consideration by combining one representative interpretation from each principle (capability, equality, and responsibility). Figure 3 illustrates this for a single illustrative country, Germany, highlighting how systematically varying the weights among the three equity interpretations translates into different 2030 reduction targets. The color gradient in the figure conveys the required emissions reductions (resulting from a weighted combination of one capability, equality and responsibility interpretation) at any given point in the triangle—the location of such points indicates via its distance from the corners by the three axes the relative weighting of the chosen C, E, and R interpretations.

Panel (A) shows the results of implementing C1-EU-GDP per capita for capability, E1-Basic-needs for equality and R1-historic emissions since 1995 for responsibility. Moving from the bottom to the top along the left leg, and thus increasing the weight of capability, increases the reduction target for Germany. The same holds when moving along the base from right to left, thus increasing the weight of responsibility (in this case the relevance of considering historic emissions). Conversely, increasing the weight of equality (i.e., moving down along the right-hand leg) reduces emission reduction targets, as an increase in the weighting of this dimension necessarily decreases the weight of the other two dimensions, each of which we find empirically for Germany to imply stronger reduction target increases. The lightness of the gradient in the bottom right-hand corner, indicating the least required emission reductions, implies that

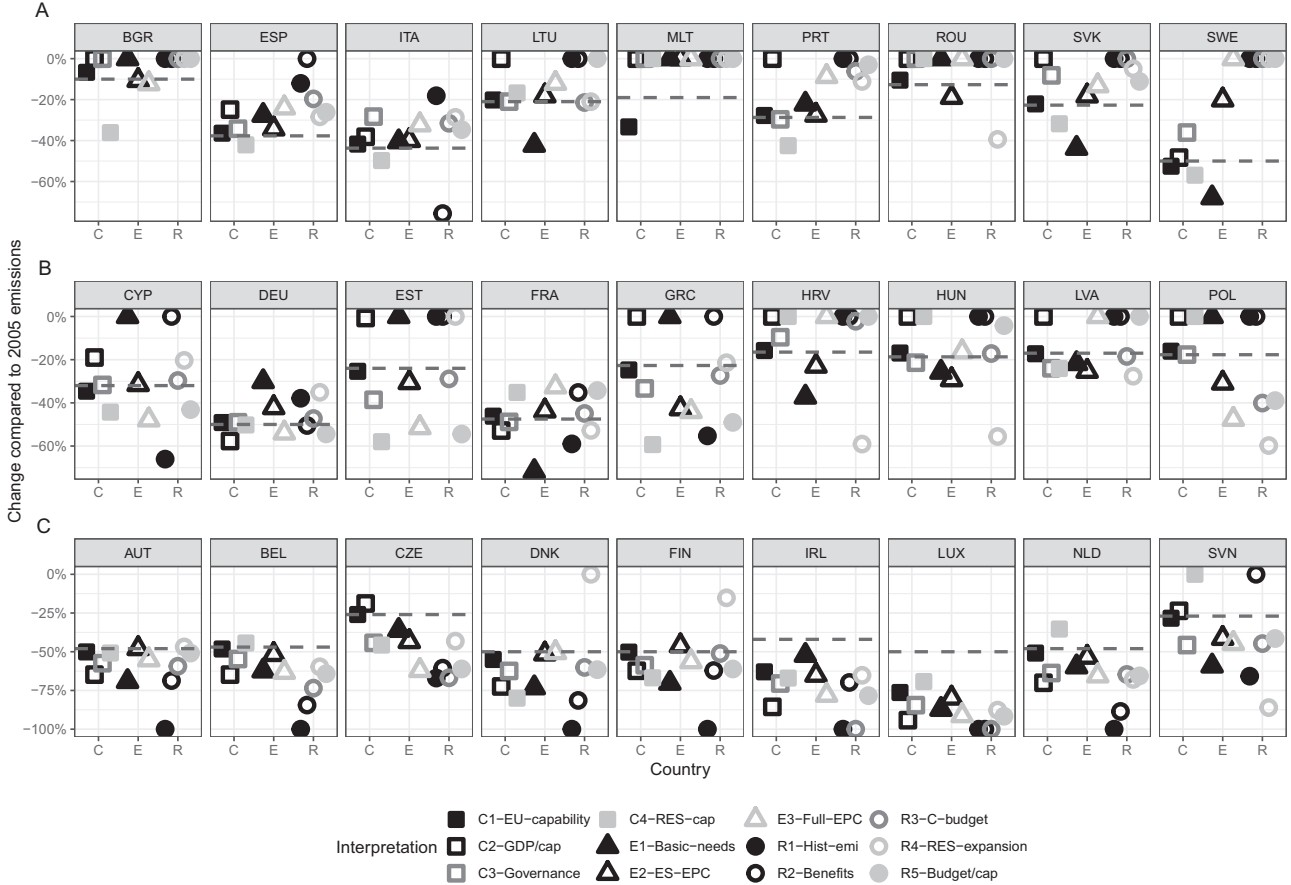

**Fig. 2 Possible country emission reduction targets by 2030 (relative to 2005) reflecting an EU-27 overall 55% reduction target (relative to 1990) for an application of each single principle (and under different interpretations of them).** The dashed line indicates the target reduction put forward in the 'Fit for 55%' proposal. The x-axis is divided into 3 categories, the first (C) contains the results of capability interpretations, (E) equality interpretations, and (R) responsibility. **A** consists of countries with allocation shares predominately above (i.e. less restrictive) than the 2021 proposal. **B** consists of countries with some scenarios leading to more stringent reductions, and others less, than compared to the 2021 EU proposal, and **C** countries with the majority of scenarios leading to more strict reductions.

avoiding consideration of both capability and responsibility principles reduces emission reductions to a minimum.

If, however, equality refers to an E3-full equal-per capita budget interpretation, as shown in panel B, combinations of the three equity interpretations that give most weight to the equality dimension (equality corner) lead to the highest emissions reduction in 2030. Shifting the weights among the three dimensions or switching the interpretation used for each corner has a different influence on emission reduction targets depending on the country and interpretations assessed; a set of ternary charts illustrating the effects of changing all interpretations for all countries can be found in the Supplementary Information, section 5, or can be generated via use of an interactive web tool developed for this framework (for details, see https://wegcenter. uni-graz.at/effort-sharing/).

Calculating the resulting emissions reductions requirement of any given combination of three interpretations for all EU countries provides a wealth of information for countries in terms of their negotiation position—considering the maximum emission points in the charts represent the upper bound of equity-compatible emissions—but also the rate of change in reduction levels as the weighted combination moves away from such a point or isoline. Combining this information on all EU Member States makes possible the identification of possible points of agreement in future effort-sharing negotiations, discussed in further depth in the next section.

**Possible negotiation convergence points**. In negotiating an effort-sharing agreement, agents (in our case EU Member States) could be motivated by a number of aims. One could be to minimize deviations from planned reductions as a result of established policies. Implementing such a target in our analysis would mean that when determining national budgets to 2030 using a single interpretation of each of the three equity principles from Table 1, a weighted combination of the three can be identified that ensures that countries have to do the least additional effort beyond what they agreed in 2018 for their respective reduction by 2030. They may want to keep planned reductions as close to this prior agreement as possible, in order to avoid sudden drastic changes in requirements or policy. In this case, a combination of interpretations can be identified which minimizes the aggregate effort of all EU countries beyond their prior agreement. Formally, the sum of squares of these deviations is minimized (see "Methods"). We define each of these weighting combinations as "negotiation points" and calculate one for each possible combination of the three equity criteria interpretations discussed (60 in all). As a second metric for comparison, we also minimize aggregate deviation from what is the upper bound of equity-compatible emissions for each country, i.e. an emission level that can be considered equitable by at least one interpretation. The results of these calculations for both cases, namely, a 'least deviation from past share allocation' (blue points, corresponding to minimizing the deviation compared to the 2018 ESR) and a

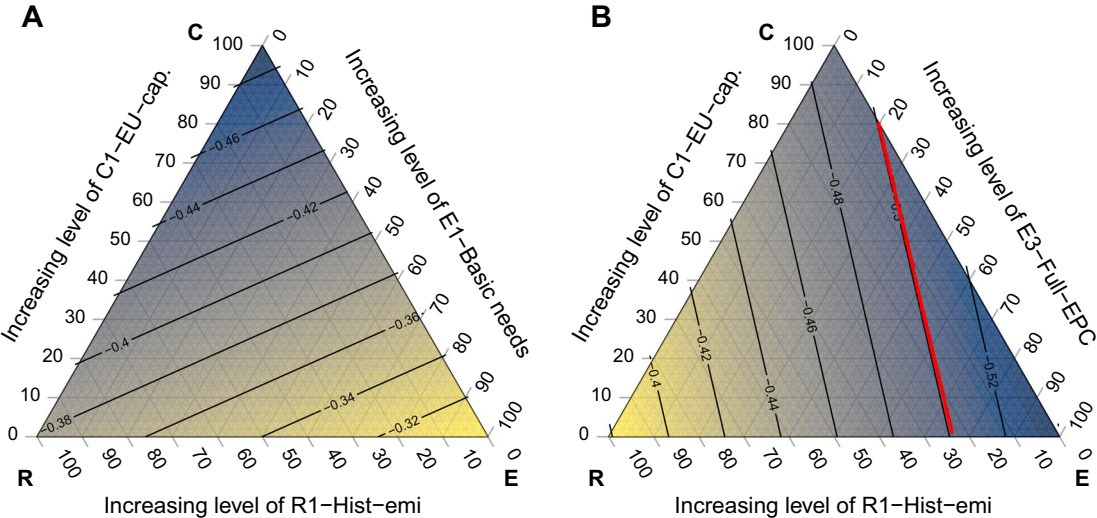

**Fig. 3 Equity triangles illustrating the required emission reductions for Germany arising from combinations of equity interpretations, where the weights of the capability, responsibility, and equality interpretations sum to 1.** The isolines indicate emission reduction targets (by 2030, relative to 2005) and are labeled accordingly. For each point on an individual ternary chart, the color indicates the level of emissions reductions required, with yellow corresponding to lower levels, and blue higher. The level at each point is the result of a weighted combination of the three equity interpretations indicated on the chart axes. As an example, a point in the middle of **A** is the reduction amount given an equal combination of C1-EU-capability, E1-Basic-needs, and R1-Hist-emi. The red lines indicate the EU suggested reduction level in the Fit for 55 proposal—for Germany, −50%—if the value falls within the range of the chart. In **B** a different equality interpretation is applied (E3- Full-EPC).

'least deviation from the upper bound of equity-compatible emissions' (marked in orange), are identified in the ternary inset panel in Fig. 4. Note that results of the latter are robust against the integration of any further equity interpretations as long as our interpretations cover the overall possible range, a goal which guided their selection (see SI, sections 2–4). Each point in the inset panel represents a combination of one each of an equality, responsibility, and capability interpretation (applied to all countries) which meets the 55% EU reduction target.

The ternary subpanel of Fig. 4 shows that these negotiation convergence points span the negotiation space, indicating a variety of combination weightings that could likely result from negotiations if Member States follow this rationale. Some trends do emerge. Comparing a minimization of effort from the 2018 ESR to the alternative of maximal equity-compatible emissions, we find the ESR-based negotiation points exhibit much more clustering (as point size indicates frequency), most agreement points are either almost fully capability-weighted, or roughly 50% capability, and the rest either equality or (to a lesser extent) responsibility interpretations. As one of our four capability interpretations is based heavily on the 2018 ESR (and a second based purely on GDP per capita is highly correlated; see "Methods"), the clustering near a 100% capability allocation is not surprising. However, even when minimizing aggregate EU Member State deviation from maximum equity-compatible emissions, a similar clustering occurs, although less strongly. Regardless of the minimization criteria or equity interpretation, what is consistent is the presence of capability in practically all negotiation points; the only equity criteria to do so. The negotiation points seem to explain the current EU negotiations, as these results would make it seem unsurprising that the 2021 proposal emphasizes a GDP per capita-based allocation (and thus, the capability dimension). This analysis thus has identified why a capability principle interpretation based on GD/capita ranks so prominently within the EU negotiation process.

While the negotiation points results do acknowledge the relevance of a capability interpretation, they also emphasize the possibility for a number of other negotiation points (e.g., points of

agreement or compromise) that incorporate other aspects of fairness. These points thus represent solutions which may be on the whole less burdensome for the EU to adopt in terms of emissions reductions compared to previous agreements and may at the same time increase buy-in from countries that up to now may not have agreed with an approach emphasizing capability as the only relevant factor in budget allocation.

Wherever they fall in the ternary chart, these negotiation points imply a weighted combination of three equity interpretations leading to an allocation across EU Member States, which is illustrated in the main panel of Fig. 4. The plot indicates the range of the 2030 reduction targets across all negotiation convergence settings, i.e., the range and extremes of country reduction targets that results across all negotiation convergence points for both the "closest to upper bound of equity-compatible emissions" and "minimal change from the 2018 ESR" on a per capita basis. The distributions of these results can be compared to the solid black line, indicating EU Member State 2030 targets as proposed by the EC in 2021. When comparing to this effort-sharing proposal, we find that a broader acknowledgment of equity dimensions enhances the variability of emission reduction targets. Countries that have the lowest emission reduction obligations under the EU proposal tend to have even less restrictive ones when further equity dimensions are considered as well (e.g. Bulgaria, Romania, Latvia, Croatia). Conversely, countries with the highest reduction obligation under the EU proposal tend to have even more stringent ones once one or both of the other equity dimensions are considered as well (e.g. Austria, Denmark, Netherlands, Finland). The two notable exceptions to the latter group are Germany and Sweden, who rank high in EU proposal obligations, but would not have to increase their reduction target under alternative considerations, most importantly due to their recent strong emission reduction, implying for example no additional reduction obligation from historic emissions consideration.

Figure 4 further indicates that minimal deviation from the earlier negotiation result (ESR 2018) for most countries overall comes close to what would have resulted from a focus on remaining closest to the upper bound of equitable-compatible

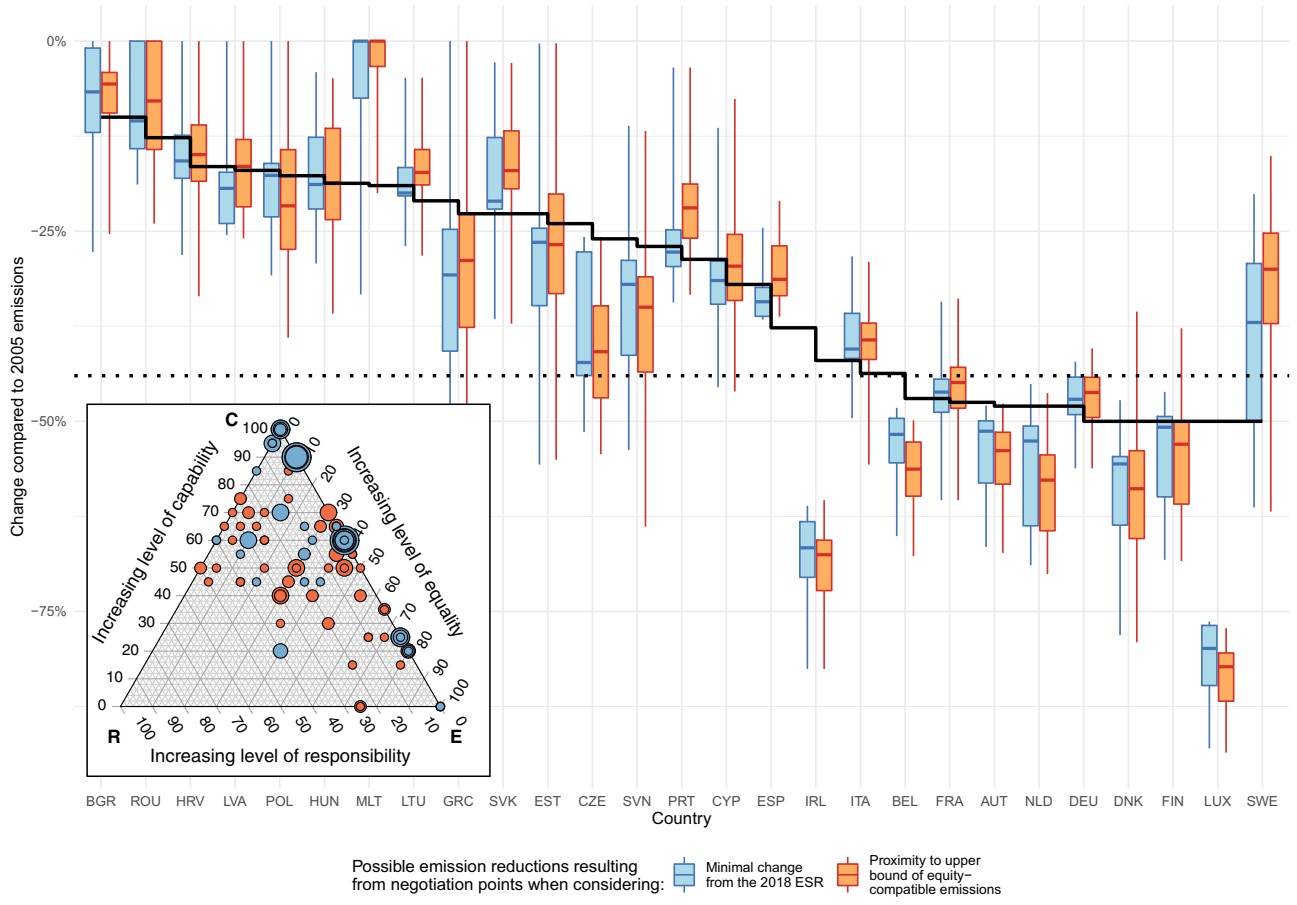

**Fig. 4 Negotiation points derived from optimal weighting of equity interpretations and their resulting reductions by 2030 by country.** The box and whisker plots in the main panel show the resulting range of emissions reductions by 2030 by country, corresponding to the negotiation convergence points shown in the inset equity triangle. The solid black line indicates the 'fit for 55%' reduction target for each country and the dotted line the required reduction if each country were to reduce an equal percentage in the effort-sharing sector to meet the 2030 goals (corresponding to −44%). Points in the inset ternary chart denote negotiation points when using an allocation approach comprised of weighted combinations of three equity principles, point size indicates frequency, i.e. multiple points at one location. Orange points and boxes are for minimizations of EU aggregate additional effort above Member States upper bound of equity-compatible emissions, and blue for minimal deviation from the 2018 ESR.

emissions—there is little deviation between the blue ESR ranges and orange upper-bound ones. However, a few countries see larger deviations between the two approaches, indicating that e.g. Sweden and Portugal took on (slightly) stronger reductions in the ESR 2018.

## Discussion

The paper introduces a systematic and transparent approach to evaluate national emissions reduction efforts according to different equity dimensions. We introduce a number of different equity interpretations and propose a method to assess possible convergence points that would minimize country effort deviations from different yardsticks and thus help to identify acceptable negotiation outcomes.

Applying this approach to the effort-sharing negotiations of the 2030 climate ambition across EU 27 Member States, which will continue in 2022, we find that the 2021 EU proposal is for many countries consistent with our 2030 emissions estimates (from the negotiation convergence points).

The GDP-per-capita-based capability approach of the EU thus captures the dynamics of the majority of countries well. The dimensions of equality and responsibility, however, are equally important to consider. Introducing them increases the divergence of country reduction targets; countries with the lowest emission reduction targets resulting from capability interpretations alone,

such as Bulgaria and Romania, tend to have even lower targets when using a combined approach, thus allowing for reducing their emission reductions for 2030. The converse is true for countries with high reductions from only a capability approach; considering other equity dimensions would lead to considerably higher burdens. This holds largely independently of which interpretations are employed for these two further dimensions.

Our results also can be read as one explanation of why the EU has preferred the capability approach to inform its effort-sharing allocation. When EU Member States seek to maximize emission allowances within an equity-compatible range, particularly if only one equity dimension is desired to minimize complexity, the capability dimension emerges as the indicator of choice—across all the potential negotiation space it has by far the highest weight among all dimensions and even when varying across all interpretations. And this is exactly what the EU has done in its 2018 effort-sharing regulation and the EC has again implemented in its 2021 proposal.

However, as we have shown, simple, transparent and systematic approaches to incorporate additional considerations are within reach, and, more importantly, can produce allocations that could lead to more buy-in than the 2021 single-indicator-oriented approach. To further support future negotiations and increase accessibility of our framework, we have developed an online tool that can be used to visualize all results discussed here (in the

context of the EU negotiations) and investigate the implications of user-defined weightings of all interpretations on eventual emissions allocations. The tool can be accessed online at https://wegcenter.uni-graz.at/effort-sharing/.

We note that the results of our approach can be employed to inform parties in ongoing political processes but do not serve as a projected end-point for negotiations, due to a number of factors. Our aim is to provide a framework for distributing future emissions budgets based on well-established equity principles, but it must be emphasized that contextual factors, the choice of interpretations, and their implementation can lead to differing outcomes. Numerous potential proxies could be suggested for a given interpretation (see Supplementary Information section 2 for discussion on possible governance indicators). Application of increasing renewable shares, as either a responsibility or capability interpretation, is an apt example; valid arguments can be made to place it under either equity consideration, but arguments against are also relevant. Implementation of an RES-based capability interpretation is problematic as it can be seen to reward countries for a lack of past effort without encouraging lagging countries to act by increasing their reductions. Also, past performance regarding RES is not a reliable indicator of future performance, as political or economic conditions may lead to changing capability. For a further discussion on the ambiguity of renewables as an equity interpretation, see the Supplementary Information, section 4.

Beyond choice of an interpretation, its application can have varying effects on outcomes. The most obvious example is the question of when to start taking historical emissions into account; we choose 1995 based on publication of the IPCC's Second Assessment Report. However, valid arguments can be made that other, particular earlier, years should be chosen. We find that while that is the case, shifting the year has less of an effect on outcomes compared to the initial choice to consider countries' historical responsibility. Overall emissions reductions for countries are mostly unaffected or would see only minimal changes (see Supplementary Information, section 3 and Supplementary Fig 2 for further detail).

For the specific case of the EU, both the historical emissions and inherited benefits (R1 or R2) interpretations acknowledge the specific context at the time, with Eastern European countries having been comparatively emission intensive at lower efficiency up to 1990 with emissions plummeting thereafter. Historic emissions consider aggregate emissions over the whole period back to 1995, but not including the high-emissions period up to 1990. Benefits received are derived from capital stock available in 1995 (i.e., after economic restructuring), and are evaluated using a recent average EU emission intensity (and not the historical—and more emission intensive—levels from the years before 1990).

Approaches such as presented here also need to be embedded in larger policymaking contexts; the communication of the EU Green Deal emphasizes that the combination of the climate neutrality goal by 2050 and ambitious 2030 climate targets together act as a crucial framework to provide long-term certainty and predictability for investments[49]. Considering the 2050 perspective, we note that for some countries a small subset of negotiation points result in zero emissions reductions compared to 2005 levels, which might imply allowances for increasing emissions until 2030. While most of the countries this refers to had higher emissions in 2019 than in 2005, and thus would still be required to reduce emissions, it does not hold true for three countries (Romania, Croatia, and Greece). Here, a minority of weighting combinations might allow rising emissions. Such a development would need to be considered in the context of the EU net-zero emission target by 2050, which—given the limited potential of negative emissions—essentially translates to a close-to-net-zero target for every country. Thus, equity considerations may prohibit rising emissions up to 2030, particularly if this would imply an increase in future stranded assets, and would need focused deliberation.

Given the above issues, our approach provides a framework for discussing equity-compatible 2030 targets, but it is up to negotiators to not only choose the weighting amongst the three equity dimensions (and their respective interpretations) but also to decide how strongly an interpretation influences emissions reduction. On the latter we have consistently determined allocations based on percentage deviations of interpretations from an EU average, where applicable (i.e., for all interpretations not based on budget approaches, directly translating to emission reductions by country; see "Methods" for further details).

While we do not analyze the actual political processes, the related governance literature has informed our analysis. First, it identified the particular relevance of path dependency, i.e., the notion that policy decisions once made within a certain frame tend to stick to that frame[14,50], which guides our choice of past EU effort-sharing regulation in one approach to derive negotiation convergence points. Second, the governance literature identifies the relevance of timely transparent information. In particular, the 2019/2021 EU decision of implementing to remain within a Paris compatible carbon budget in combination with the Corona-aftermath and "building back better" may represent a critical juncture in institutional development, at which "decisions of important actors are causally decisive for the selection of one path of institutional development over other possible paths"[51], and conflict over ideas has been identified as important for institutional change[52]. The approach we present is intended to contribute to resolving conflicts over equity perspectives to allow for institutional development.

The results of our framework make transparent how different choices of equity interpretations can translate into different country contributions. The range of transparent equity considerations made explicit here allows for appreciating the positions of other countries, as well as for a common understanding of the range of outcomes, and thus can contribute to successful negotiations.

For regions without previous effort-sharing agreements to refer to, as available for the EU, such transparent and commonly available exploration of the negotiation space is likely to be an even more important ingredient in the process to agree on sharing among subsidiary entities.

## Methods

Calculation of potential country budgets to 2030 occurs in four steps. The first step is determination of the total EU budget to 2030, given an assumed reduction target compared to historical emission levels, and assumptions on the share of (non)ETS emissions in total EU emissions.

Given an EU effort-sharing sector budget to 2030, the distribution of that budget across countries can be determined based on any number of desired allocation approaches. We develop a number of interpretations, based on (and designed to address) one of the three following equity components: (i) Responsibility, (ii) Capability, and (iii) Equality. We assume that any budget distribution will take into account one interpretation from each equity component, to a varying degree, allowing for combinations of the interpretations. However, before combining them, their individual impact is calculated, as discussed in the section "Description and calculation of interpretations" below.

We first calculate a distribution of emissions as a result of each interpretation, as though the interpretation were the only factor being considered. We then combine the interpretations as described above (using a single interpretation for each equity component) in a weighted combination, with weights of the three chosen interpretations summing to 100%.

In the final step, for each of the possible combinations of interpretations of the equality, responsibility, and capability dimension (i.e. one interpretation from each equity component) we determine the weights among the three that would result in the least deviation from a given reference point. One such reference point is the set of emission reduction targets suggested in the 2018 EU Effort Sharing Regulation (meant to reflect path dependency), the other is a country's upper bound of equity-

compatible emissions (the maximum that still could be considered equitable under at least one combination of interpretations across the three dimensions). It should be noted that a common feature for all interpretations is a zero-restriction. A zero-condition is imposed, wherein a country is prohibited from having a positive change in emissions compared to 2005. If the result of a raw interpretation is a positive change, the relevant country is allotted no reduction compared to 2005, and the additional positive allowance initially allocated to it is distributed to all other countries on an equal-per-capita basis. Similarly, we assume that net-negative emissions in 2030 are infeasible, and as such set the maximum required reduction to be 100% of 2005 emissions.

Unless otherwise specified, data was obtained from the EU's EUROSTAT database[53].

**Determination of the effort-sharing emission target level in 2030.** The effort-sharing budget from 2020 to 2030 is determined via the EU target goal in 2030 of an at least 55% reduction compared to 1990 emissions.

$$G_{EU}^{2030} = \left[ \left( e_{EU}^{1990} * (1-r) \right) - \left( e_{EU}^{1990} * (1-r) \right) * ETS \right] + \Delta LULUCF \quad (1)$$

Where $G_{EU}^{2030}$ is the target maximum emissions of the EU effort-sharing sector in 2030, $e_{EU}^{1990}$ the emissions of the EU in 1990, $r$ is the total reduction percentage, ETS the ETS share of emissions, and $\Delta LULUCF$ the change in sinks due to land use, land use change and forestry when comparing 2030 to 1990. For our scenarios, we assume $r$ to be 0.55, ETS 0.37, and $\Delta LULUCF$ to equal an increase in sinks of 98.8 Mt $CO_2$, the target specified in the EU Fit for 55 climate package[9].

**Description and calculation of interpretations.** This section gives an overview of the inputs and specific calculation steps resulting in country budgets and corresponding emission reductions implied by each interpretation. We define twelve total interpretations of three equity components; five interpretations of Responsibility, four of Capability, and three of Equality (see Table 1).

For interpretations which function by changing a baseline allocation (denoted as $b_j$) depending on the distribution of a given interpretation variable (e.g. GDP per capita), we utilize a baseline that assumes an equal percentage emission reduction by all countries to reach the emission requirement for 2030 (starting from 2019 values).

The interpretations described in Table 1 are summarized and calculated as follows:

(C1- EU-capability) EU implementation approximation: The EU proposed a set of country reductions to meet the 55% target, based on GDP per capita as well as other unspecified considerations, in two formulations, a "bound" version, with reductions limited between 10% and 50%, and an "unbound" set. We approximate the influence of GDP and the previous ESR reduction targets using a linear model to provide a rough analog of the EU's capability approach in our interpretation set. The results of the model can be found in Table 2.

(C2- GDP/cap) GDP per capita: Countries with a higher GDP per capita (in 2019) are allocated a smaller emission budget, i.e. a stricter emission reduction target. For an intensity of 100% for each Member State (MS) the change in GDP/capita from the EU-27 average is translated to an equal % deviation in the effort-sharing emission reduction from the EU-average.

The process to calculate country emission shares using this interpretation is:

$$s_j = \frac{\frac{b_j}{f_j/f_{EU}}}{\sum_{j=1}^{J} \frac{1}{f_j/f_{EU}}} \quad (2)$$

**Table 2 Linear approximation of the EU's use of GDP per capita in allocating 2030 emissions reductions, including consideration of the previous 2018 Effort Sharing Regulation (ESR) reduction targets.**

|  | Dependent variable |
|---|---|
|  | Fit for 55% proposed reductions |
| 2019 GDP per capita | 0.005*** |
|  | (0.0004) |
| 2018 ESD country target | 0.638*** |
|  | (0.053) |
| Constant | 0.216*** |
|  | (0.012) |
| Observations | 27 |
| $R^2$ | 0.985 |
| Adjusted $R^2$ | 0.984 |
| Residual Std. Error | 0.022 (df = 24) |
| F Statistic | 798.251*** (df = 2; 24) |

*$p < 0.1$; **$p < 0.05$; ***$p < 0.01$.

where $s_j$ are country target emissions shares in 2030 for $j = 1,...,J$ for all $J$ EU countries; $b_j$ is the baseline emissions share (e.g. the distribution which occurs if weight is set to zero, and the distribution which is altered by the GDP per person criteria), $f_j$ the interpretation to be applied, e.g. here, country $j$'s GDP per capita, and $f_{EU}$ the average GDP per capita in the EU.

(C3- Governance) Government effectiveness: This interpretation is calculated using Eq. 2 as in C2- GDP per capita; countries with higher index rankings are allocated a smaller emission budget.

(C4- RES-cap) Renewable growth capacity: reflects the difference of the Member States in terms of their change in renewable share from 2005 to 2019 compared to the EU-27 total, calculated as a population-weighted average. Higher increases in renewables share compared to the EU-average results in higher-than-average emissions reductions in 2030.

Calculation:

$$d_j = b_j^{2019} \Bigg/ \left( \frac{RES_j^{2019} - RES_j^{2005}}{\left( \frac{\sum_{j=1}^{J} \left( \left( RES_j^{2019} - RES_j^{2005} \right) * p_j \right)}{\sum_{j=1}^{J} p_j} \right)} \right) \quad (3)$$

$$s_j = \frac{d_j}{\sum_{j=1}^{J} d_j} \quad (4)$$

where $s_j$ is the share of emissions received by country $j$ where $j = 1,...,27$ EU countries, $RES_j^y$ is the share of renewables in country $j$ in year $y$, either 2019 or 2005, and $p_j$ is population in country j.

(E1-Basic Needs): Budget allocations for the basic-needs interpretation utilize results by Rao and Min[54] and Kikstra et al.[55] on the energy requirements to meet basic needs and attain decent living standards. The work of Kikstra et al. provides estimates of country-explicit energy requirements to achieve sufficient nutrition, housing, and transportation to meet established standards of living. We utilize these estimates to allocate a portion of the EU effort-sharing budget as a priority measure to be used in meeting basic needs thresholds for the portion of the population most at risk of poverty.

Based on the EUROSTAT dataset "Persons at two-fold risk of poverty" the number of persons in a country which should be allocated a basic needs energy allotment are determined. The proportion of people living below this poverty line are pre-allocated a set amount of emissions, to be used to meet basic needs considerations from Kikstra et al. Country emissions necessary to domestically produce the energy required to fulfill the decent living standards for all those under the poverty threshold is calculated via current national emissions intensity data from the European Environment Agency[56]. Countries are in a first step given the necessary amount to cover all persons under the poverty headcount threshold for the period of 10 years. The remaining emissions are divided among countries in an equal-per-capita manner.

(E2- ES-EPC and E3- Full-EPC) Equal per capita convergence: Reflects a distribution of the budget to achieve equal-per-capita emissions in 2030. Thus, the calculation of country emission shares in 2030 is simply:

$$s_j = \frac{p_j}{\sum_{j=1}^{J} p_j} \quad (5)$$

where $s_j$ are country emissions shares for $j = 1,...,J$ for all $J$ EU countries and $p_j$ is population of country j. There are two similar interpretations; total emissions per capita and total effort-sharing sector emissions per capita. As the names imply, total emissions per capita uses the total country emissions in the calculation, whereas the latter uses only effort-sharing sector emissions.

(R3- C-budget): The total emissions budget for the ES sector, (calculated by a fictitious linear path from 2019 to 2030) is split among Member States according to population. The resulting emissions budgets produce target paths up to 2030 with a corresponding target distribution for the effort-sharing sector.

The calculation is as follows:

(1) Establishing the total budget to 2030:

$$B = \left[ \left( e_{EU}^{2019} - e_{EU}^{2030} \right) * \frac{t}{2} \right] + t * e_{EU}^{2030} \quad (6)$$

where $B$ is the total budget, $e_{EU}^y$ the effort-sharing emissions of the EU in year $y = 2019$ or 2030 (the target emissions in the case of 2030) and $t = (t_{end} - t_{start}) + 1$ with $t_{end}$ and $t_{start}$ the ending or starting years in the budget calculation, 2030 and 2019.

(2) Calculation of raw country budgets:

$$e_j^{2030} = B * \left( \frac{p_j^{2019}}{p_{EU}^{2019}} \right) * \frac{2}{t} - e_j^{2019} \quad (7)$$

where $e_j^y$ represents emissions in country $j$ in either year $y = 2030$ or 2019 depending on the superscript. $t$ is again the simplification of $\left( (t_{end} - t_{start}) + 1 \right)$ as in step 1 above, indicating the years between the start and end points of the budget calculation (12), and $p_j^{2019}$ and $p_{EU}^{2019}$ representing 2019 populations of country $j$ and the entire EU, respectively. While some approaches utilizing population in emissions allocation algorithms also use projections of future population (see, for example, discussion in Williges et al.[57]) due to the short timescale involved (10 years) and the relative projected stability of EU Member State populations over that time period – with most remaining within a few percent of their current levels – we choose to utilize only current population in our algorithm.

(3) Elimination of negative budgets: To avoid the imposition of net-negative emissions on countries, any negative emissions as a result of the interpretation are removed. The country in question is allotted a reduction of 100% compared to 2005 values, and the additional reduction needed to meet budget goals is instead equally distributed to countries not experiencing net-negative emissions.

(4) Calculation of the distribution of emissions to each country:

$$s_j = \frac{e_j^{2030}}{G_{EU}^{2030}} \tag{8}$$

where $G_{EU}^{2030}$ is the total emissions of the EU for the year 2030, allowing for calculation of the share of country emissions in 2030.

(*R1- Hist-emi*) Historical emissions from 1995: reflects the use of fossil fuels since the year 1995. The point of time at which the remaining budget is allocated on an equal-per capita basis is shifted back to 1995.

The calculation follows similar steps as the R3- C-budget approach above, but with some changes to steps 1 and 2, as follows:

(1) Establishing the total budget from 1995 to 2030:

$$B = \left( \left( e_{EU}^{2019} - e_{EU}^{2030} \right) * \frac{t}{2} \right) + t * e_{EU}^{2030} + e_{EU}^{1995-2019} \tag{9}$$

where $B$ is the total budget, $e_{EU}^{y}$ the effort-sharing emissions of the EU in either year (*y*) 2019 or 2030 (the target emissions in this case), with $e_{EU}^{1995-2019}$ the total EU emissions from the year 1995 to 2019 and $t$ equal to $\left( (t_{end} - t_{start}) + 1 \right)$, the number of years between the start and end points in the budget calculation, in this case, 12.

(2) Calculate raw country budgets:

$$e_j^{2030} = \left( B * \left( \frac{p_j^{2019}}{p_{EU}^{2019}} \right) - e_j^{1995-2019} \right) * \frac{2}{t} - e_j^{2019} \tag{10}$$

Of note here – differing from the R3- C-budget calculation steps – is the removal of individual country emissions in the historical period from the calculation ($e_j^{1995-2019}$). Again, $p_j^{2019}$ and $p_{EU}^{2019}$ represent 2019 populations in individual countries and the entire EU, respectively (regarding on our choice to utilize 2019 population only, see discussion in the Methods description of interpretation *R3*).

From this point, the calculations follow steps 3 and 4 (Eq. 8) in the derivation of R3- C-budget.

(*R2-Benefits*) Inherited benefits of emissions: incorporates the benefits a country has obtained due to emissions prior to the year 1995, interpreted here as being the embodied emissions in national capital stock. Using capital stock estimates, GHG budgets from 2019 to 2030 are scaled identically to the past emissions consideration above, but here based on pre-1995 emissions embodied in each country´s capital stock in 1995.

Calculation steps:

(1) Establishing the total budget to 2030:

$$B = \left( \left( e_{EU}^{2019} - e_{EU}^{2030} \right) * \frac{t}{2} \right) + t * e_{EU}^{2030} + kB_{EU} \tag{11}$$

where $B$ is the total budget, $e_{EU}^{y}$ the effort-sharing emissions of the EU in year $y =$ 2019 or 2030 (the target emissions in this case) and $t$ the number of years between the start and end points in the budget calculation $\left( (t_{end} - t_{start}) + 1 \right)$, in this case, 12. The variable $kB_{EU}$ represents the total emissions embodied in capital stock (a proxy for inherited benefits) calculated for the EU, based on Williges et al.[57].

(2) Calculation of raw country budgets:

$$e_j^{2030} = \left( B * \left( \frac{p_j^{2019}}{p_{EU}^{2019}} \right) - kB_j \right) * \frac{2}{t} - e_j^{2019} \tag{12}$$

Similarly to Step 2 in the calculation of *R1-Hist-emi*, individual country estimates of inherited benefits through capital stock in terms of according embodied emissions ($kB_j$) are removed from total budget allocations.

From this point, the calculations follow steps 3 and 4 (Eq. 8) in the derivation of R3- C-budget.

(*R4- RES*) Renewables implementation: reflects the difference of the Member States in terms of their change in renewable share from 2005 to 2019 compared to the EU-27 total, calculated as a population-weighted average.

Calculation:

$$d_j = b_j^{2019} * \frac{RES_j^{2019} - RES_j^{2005}}{\frac{\sum_{j=1}^{J} \left( \left( RES_j^{2019} - RES_j^{2005} \right) * p_j \right)}{\sum_{j=1}^{J} p_j}} \tag{13}$$

$$s_j = \frac{d_j}{\sum_{j=1}^{J} d_j} \tag{14}$$

where $s_j$ is the share of emissions received by country $j$ where $j = 1,...,27$ EU countries, $RES_j^{y}$ is the share of renewables in country $j$ in year $y$, either 2019 or 2005, and $p_j$ is population in country j.

(*R5- Cumulative emi/cap*) Historical cumulative emissions per capita:

$$s_j = b_j / \frac{\left( \frac{e_j^{1995-2019}}{p_j^{1995-2019}} \right)}{\left( \frac{e_{EU}^{1995-2019}}{p_{EU}^{1995-2019}} \right)} \tag{15}$$

As in other interpretations, $s_j$ represents the share of emissions of country $j$ in 2030, $b_j$ refers to the baseline emissions share, $p_j^{y}$ the country population, summed for years (*y*) between 1995 and 2019, and $e_j^{y}$ country emissions, again summed for years between 1995 and 2019, for either individual countries (subscript *j*) or the EU (subscript EU).

**Combining interpretations into responsibility, capability, and equality framing**. Using the interpretations listed in the previous sections, an allocation that combines elements of capability, responsibility and equality principles can be generated, as in Fig. 1. We calculate a weighted combination of three interpretations (one from each equity cornerstone) with the sum of the weights of the three interpretations equal to 1. These weighted combinations are used to generate the ternary charts found in Fig. 3; further charts for all countries can be found in the Supplementary Information, section 5.

**Calculation of negotiation convergence points**. In addition to country emission budgets when applying different equity interpretations and weights, we calculate the combination of equality, responsibility, and capability weightings which minimize (a) the sum of squared changes in per capita country budgets from their maximum possible allowance to the commonly-weighted level, or (b) the sum of squared changes from the original 2018 ESR agreement to the common weighting. This minimization is calculated for all potential combinations of equity interpretations (4 capability × 3 equality × 5 responsibility = 60 combinations).

As shown below, the goal is to minimize the total (over all countries) squared percentage difference between an individual country maximum preference and the interpretation which results due to a common weighting of the three cornerstones.

$$\text{minimize} \sum_{j=1}^{J} \left( \frac{a_j^{max} - a_j(h,c,q)}{a_j^{max}} \right)^2 \tag{16}$$

$$\text{subject to } h + c + q = 1$$

where $a_j^{max}$ are the maximum per capita country emissions allowances given across all possible interpretations under consideration for $j = 1,...,J$ for all $J$ EU countries (or the allowances according to the EU Effort Sharing Regulation of 2018, respectively), and $a_j(h,c,q)$ indicates that country emissions are a function of the weights for $h$ (historical) responsibility, $c$ capability, and $q$ equality weightings. Note that the three interpretation weights must add to one, i.e. the allocation is fully qualified.

## Data availability
The country emission budget allocation and reduction target results generated in this study have been deposited in a permanent public Github repository linked to Zenodo, accessible here: (https://doi.org/10.5281/zenodo.6574309)[58]. Source data for all figures found in this work can also be obtained from the same repository. All source data for calculations are freely available from the EUROSTAT database (which can be found at: https://ec.europa.eu/eurostat/data/database), with the exception of data on energy requirements for basic needs calculations, which were obtained from the authors of Kikstra et al.[55] upon request.

## Code availability
All code used to generate the results and figures discussed in this paper and Supplementary Information file can be found in a permanent public Github repository linked to Zenodo[58], accessible here: (https://doi.org/10.5281/zenodo.6574309).

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

## Acknowledgements
The authors thank Jarmo Kikstra for cooperation in operationalizing basic needs considerations, Elina Brutschin for operationalizing governance indicators. This research has received financial support from the Austrian Federal Ministry for Climate Action (grant 2020-0.513.449 "Updating European NDCs": K.W.S, K.W., L.H.M, F.M. and K.R.), the Austrian Science Fund FWF (project P 33169 "Basic Needs and International Climate Justice": K.W. and L.H.M), the European Union's Horizon 2020 research and innovation programme (Grant Agreement No. 837089 "Sustainable Energy Transition Laboratory": K.W.S. and K.W.), and the University of Graz (K.W.S., K.W. and L.H.M)

## Author contributions
The authors confirm contribution to the study as follows: study conception and design: K.W.S, K.W., L.H.M., and K.R.; data collection and model generation: K.W. and F.M., analysis and interpretation of results: K.W., K.W.S., L.H.M., K.R., draft manuscript preparation: K.W.S, K.W., L.H.M. All authors K.W.S, K.W., L.H.M., F.M. and K.R. reviewed and approved the final version of the manuscript.

## Competing interests
The authors declare no competing interests.
