## [Peer Review File · Nature Communications]

Reviewer comments, first round of review

Reviewer #1 (Remarks to the Author):

The article 'Sharing the effort of the European Green Deal among countries' proposes a new analytical framework to assess and distribute emission reduction commitments among several actors based on three justice principles, namely capability, equality and responsibility.

In general, the proposed framework (though I am not an expert on the applied methods) is interesting and largely convincing. However, I do have some suggestions for improvement.

1. My main point is that the conclusion and discussion of the political implications of the framework is a bit short and under-complex. In the conclusions, the authors argue that the framework could help to identify acceptable negotiation outcomes. While this might be true, the political negotiations within in the EU are not only driven by considerations of fairness/economics. Additionally, they are not confined to the climate realm. Meaning that the commitments of individual states are influenced by side-deals, historical path dependencies, power struggles, different cultures, technological capabilities, domestic politics etc. Therefore, even though the proposed framework might be quite straightforward in purely rationalist/economic terms, it remains to be seen whether it really would make a difference in real-life political negotiations. For instance, if some countries would see a chance to lower their ambition based on some of the proposed mechanisms (Bulgaria, Romania), others would certainly object to using these particular mechanisms. Equally, if some countries would have to increase their commitments even though they already have the highest commitments, they would most likely try to block frameworks that use these mechanisms. Eventually, opening up the currently agreed upon mechanisms (mostly capabilities), could lead to cherry picking of mechanisms by the countries and seriously complicate or derail the already fragile EU internal climate negotiations.

In my opinion, the article should at least discuss these limitations in the conclusions. It could also be interesting to discuss some of these limitations in more detail in relation to an exemplary country (e.g. the authors could add that to the section on Germany). It would be even more interesting to compare the opportunities and possible hindering circumstances in two different states, preferably one where a different allocation between the mechanisms would lead to an increase in commitments and one where it would allow for a reduction.

2. The authors base their three effort sharing mechanisms on three well-known, quite broad justice principles. However, they then narrow them down to specific variants, for instance the capabilities mechanism to the ability to pay principle and then to the GDP of individual actors. One could argue that there are different operationalisations of the capability principle e.g. by Schlosberg and others, that do not necessarily look at the GDP but rather on specific capabilities of countries to contribute to climate abatement and climate justice. These do not necessarily have to be linked to the GDP but could consist in organizational, diplomatic, or technological capabilities. This again boils down, to the above-mentioned tendency of the article to treat countries as rationalist like-units, which could at least be discussed more openly.

3. The claim that states were only liable to know about the harmful consequences of GHGs from 1995 can of course be disputed, they already knew from the late 1980s or at least since the 1992 UNFCCC with a considerable level of certainty.

4. Some of the figures are quite hard to read, especially if one reads the paper in B/W.

Reviewer #2 (Remarks to the Author):

Review of 'Sharing the effort of the European Green Deal among countries'

Review by: Claire Dupont

I was very pleased to have the opportunity to review this paper. It covers a timely topic and presents a considered framework for assessing how to think about sharing the effort in meeting the 2030 greenhouse gas emissions reduction target in the EU. I was therefore enthusiastic when I received the invitation to review.

I would first like to commend the authors on their paper. Their efforts to present an approach show the complexity in considering questions such as 'fair share' or 'effort sharing' and the distribution of such efforts in theory and in practice. I am not an expert in previous studies on effort sharing distribution calculations, and I learned a lot in reading this paper.

While being enthusiastic about the topic and approach of the paper, I was, however, left dissatisfied when I came to the final page of the paper. Perhaps since I am rather a researcher of climate policy and governance in the EU more broadly, I felt some of the weaknesses of the approach and the gaps in the discussion flow from the too-tight connection only to effort sharing and a focus on the 2030 timeframe. From this sense of dissatisfaction as a reader, I offer my feedback and comments in the hope that the authors will find them useful or insightful. At the same time, I acknowledge that some of my comments may at times seem to the authors to be outside the scope of what they are trying to achieve in the paper, but I would offer that as my first comment: perhaps it would be worth reflecting on the scope – what is the objective of the paper and can it be achieved within the scope set by the authors?

I list the comments in no particular order below. In summary, the comments refer to: 1) framing of the paper; 2) choices in the approach and interpretations; 3) discussion and conclusions.

1) Framing of the arguments, assumptions, objectives and discussions in the paper

Here, I highlight some of questions that came to me as I read the paper's introduction, discussion and conclusions, and how these questions also connected to the approach put forward in the paper.

- In both the opening of the article and in the concluding remarks, the authors refer to the EU process of dividing up the effort to reduce GHG emissions in sectors not covered by the ETS as 'ad hoc' and the result of 'political processes'. Indeed, I agree that the outcomes are the result of political processes, but I would not necessarily consider these outcomes to be 'ad hoc'. At the same time, I think the authors themselves (given their results) would agree that these are outcomes based within a certain proposed framing that draws upon a logic of capability, and not from an ad hoc political process. Furthermore, if the authors were to step outside their scientific circle, they could learn quite something about how EU political processes are themselves often less than ad hoc. In EU governance studies, we find research highlighting the importance of path dependency: the notion that policy decisions once made within a certain frame tend to stick to that frame. So, if a capability focus is the basis for effort sharing (or even earlier, burden sharing) decisions in the past, then it is likely that it would remain in the present/future unless major events or ideas (critical junctures, crises, e.g.) intervene to shift the policy development framework onto another path. This is best laid down theoretically by researchers on historical institutionalism, and also perhaps discursive institutionalism (Capoccia, 2016; Hall & Taylor, 1996; Peters et al., 2005; Schmidt, 2010).

- Furthermore, continuing the discussion of EU political processes, there is an implicit assumption in the approach outlined in the paper that member state preferences would be to align their negotiation position to achieve minimal effort (from line 276 and in figure 4). This implies an assumption that decision-making in the EU tends towards what scholars would call a 'lowest common denominator' outcome. Certainly, in crisis situations there seems to be evidence that the EU advances through such lowest common denominator decisions, but that is not necessarily a rule. In fact, much research on the nature of EU political and policymaking processes (when not in crisis response decision-making mode) has shown that these processes tend to lead to outcomes that are higher than the lowest common denominator and tend towards an advancement of what can be called the 'community interest' (Jones et al., 2016; Rhinard, 2019; Zaun, 2016). Explanations for why the EU has regularly adopted policies beyond the lowest common denominator can again be found in the institutional set-up and institutionalist perspectives

described above, but not only. The role of expert groups, the deliberative nature of the decision-making procedures (which tend to be set aside or at least less central in crisis situations and hence why crisis decision-making may display more lowest common denominator outcomes than normal procedures), notions of policy interconnections or spillovers, and the creeping competence of the EU level (more assertiveness and power to the European Parliament and/or European Commission) through various treaty amendments and functional policy advancement can play a role. From the perspective of the European Green Deal, I also would like to highlight the importance of the EU's policy mix approach to reaching an overarching objective. By agreeing on an overarching objective, and by negotiating on a package of measures that together aim for that overarching goal, policy makers are both constrained and free to negotiate their preferences. First, they are constrained because they will have to contribute to the overarching goal, to which they also have agreed. Second, they are constrained and freed by the possibility to choose their battles and engage in bargaining across policy files (Boasson & Wettestad, 2013; Dupont, 2016; Jordan & Matt, 2014; Mavrot et al., 2018).

- A last point on context: it seems important to consider the context for the effort sharing agreement towards 2030 within the overarching 2050 climate neutrality goal. The 2030 goals serve as stepping-stones or as intermediate goals towards the 2050 goal of achieving climate neutrality in the EU. Further, the EU accepts the scientific advice of the IPCC that the transition to climate neutrality needs to be under way within the decade to 2030. Given this overall goal and the broader context, the effort sharing negotiations also should be seen within this. There is therefore a temporal consideration to think of in the effort sharing discussion: if we think of the 2030 goal as a stepping stone towards the 2050 goal, it only makes sense that all member states make some effort in this decade as part of the process towards 2050. I think this broader context of the longer-term goal should be considered also. Would it not lead to some adjustment in the interpretations (e.g. a baseline of 'fairness' would be that all contribute at least something to emissions reductions in areas outside ETS to ensure they start on the path toward climate neutrality in a timely enough manner)? The EU has struggled at implementing long-term thinking into its policy developments in the past, but there have been some improvements in this regard (Gheuens & Oberthür, 2021; Siddi, 2021).

- I also wonder about the overall objective of the paper. It doesn't seem fully clear, yet. Is the purpose to figure out if the approach taken in the EU (focus on capability) is 'correct' or 'fair'? Is the purpose to check if an alternative approach would be 'fairer'? Is the purpose to reveal potential negotiating stances based on alternative approaches and prepare for that? And (how) does this contribute rather to scientific knowledge, and not only to the policy context at the moment? All of these may seem valid, but it is not really clear yet what the precise objective here is.

2) Choices in the approach and interpretations

Now, I do of course understand that a discussion of the political and policy processes of the EU is outside the scope of this paper. At the same time, I think it is important that the authors realise that some of the messages or assumptions that seem to exist under their choices should be better justified for the reader. In particular, when it comes to the choice of interpretations of the three main considerations – capability, responsibility and equality – the fact that the authors do not place these within the context of the EU decision-making processes or the broader context of the history of EU effort sharing decisions within a policy package approach raises some questions.

First a general note: the authors mention that their approach focuses on different equity dimensions, but they write about equality rather than equity throughout their methods, results etc. Equity and equality are not the same, perhaps this warrants some reflection.

On the choices, let me elaborate with some comments on the three considerations.

- Responsibility: Because political and policy processes in the EU are embedded within a historical and institutional context, some of the interpretations presented under the 'responsibility' consideration raised my eyebrows. First, I do wonder at the wisdom of choosing 1995 as a starting point for which countries should be liable for their emissions because of their knowledge at this time. In 1995, the EU enlarged from 12 to 15 members with Austria, Finland and Sweden joining that year. The EU was therefore a western European bloc at this time, and in the international climate negotiations, those countries that had emerged from the USSR had a special status as

'economies in transition'. The GHG emissions in these countries plummeted in the early 1990s, because of economic crisis. The understanding of common but differentiated responsibilities and capabilities at this time (up to and including the agreement on the Kyoto Protocol in 1997 and its implementation to 2008-2012) was that those global industrialised nations that were deemed to have most benefited from and contributed most to historical GHG emissions were most responsible for the problem and the solution. In 1995, economies in transition and developing countries were considered justified in their continued GHG emissions because of pressing economic development needs – the industrialised countries were supposed to take on the bulk of the effort. So, how can a 1995 start date for liability apply to Romania at the same time as to the Netherlands? This was not the context of the time. Of course, industrialised countries failed, even if they did take on some effort. Second, and here is where the interpretation on renewables also seems somewhat strange to me, when member states made some effort to expand their renewables sector, in the authors' approach, this leads to an interpretation that rewards them by adjusting their share of effort downwards. But industrialised nations generally got a head start in this expansion because of their same (failed) commitment to take action on climate change so that the economies in transition didn't have to. You could just as easily frame renewables expansion as a benefit from transitioning away from GHG emissions, or as a first-mover advantage, rather than focusing on the benefits only of the GHG emissions themselves. Lastly on responsibility, I do not know how or even whether it is advisable to think about this, but responsibility for emissions today is calculated based on the geographic location where emissions are produced. However, there are good arguments for considering historical emissions based on where the emissions are consumed. There is still a division in the EU among member states in this regard. Even if you cannot or would not bring in this dimension when it comes to considering responsibility, I do think it warrants a sentence of two of explanation. Some references on the historical context for international climate governance (Gupta, 2010; Oberthür, 1999).

- Equality: Next, let's talk about the equality consideration. I appreciated the different interpretations here, and I felt that the authors have done a good job of trying to operationalise these interpretations. I was struck in the paper by the comment (lines 74-75) that 'according to the equality principle everyone should be able to enjoy a level of wellbeing above the level required to secure basic needs', and yet the interpretation as presented then focuses on achieving basic needs (not above basic needs). In the methods section, I learned that this is understood to mean sufficient nutrition, housing and transportation to meet established standards of living (lines 453-454), which is seen as in line with the sufficiency principle: is this equivalent then to 'reaching the poverty line', which is considered the proxy (line 77)? Because this still doesn't seem to meet the 'above' basic needs level, and so somehow misses the 'equality principle' benchmark. Could you explain better why you make this choice? What would it change to really implement the equality principle along the above basic needs interpretation?

- Capability: Finally, on capability, I understand also the choices made here in the interpretations of capability as 'ability to pay'. At the same time, I do also see a missed opportunity for the authors to conceptualise, consider and perhaps even critique their own work based on the interactions among the three considerations. For example, the interpretation of responsibility that includes RES expansion as an opportunity to reward states by allowing them a lower reduction target also could be considered differently: as historical evidence of their technical capability to reduce emissions (perhaps because of technical know-how or their geographical, territorial positioning or climate that means they have a high potential for RES on their territory). Interactions among variables is not just combining them, but it is investigating how, whether, why some more have more influence in some circumstances, what sort of interaction exists, whether the interaction bolsters or negates other interpretations etc.

3) Discussion and conclusions

Having had a look at some of the assumptions of the interpretations and the choices behind the interpretations, that could often do with more explanation, justification or perhaps reflection, let's turn to the results, discussions and conclusions.

- The results show that the approach proposed by the European Commission to allocate effort sharing is probably the most sensible (focus on capability as ability to pay), because of the variation when other interpretations come in. This is already a worthy finding, and perhaps could be highlighted further. Although, because the overall objective of the paper is perhaps still unclear, it is also unclear whether or not the paper aimed to assess the European Commission's proposed approach for how sensible it is as an approach... I do feel that this is already an interesting,

welcome and important finding and I also feel that the authors may not necessarily be happy that this is what they found (am I reading too much between the lines?) since this finding is hardly highlighted.

- I would find the results easier to understand if the variations of potential effort sharing per country were put into the context of the 2030 goal. I.e. what combination of interpretations per member state adds up to the 2030 overarching goal? The authors discuss this somewhat in the text, try to discuss this a bit within the negotiation stances part, but a selection of (graphically-represented) options would be welcome. If the 2030 effort sharing goal is to be achieved, what combinations of interpretations per member state can work? This, of course, if the interpretations are more fully justified/explained as discussed already in the comments above. Also, there is a question that I feel the authors don't necessarily answer well – can these three interpretations be applied together? The variation in figure 2 is very large and begs this question. If the authors could lay out some scenarios for how these three approaches could/should be implemented together towards the 2030 goal, taking account of various other context considerations, that would be a helpful way of tying several loose ends together.

- All the rest of the discussions of the results seems to fall within the realm of 'political processes', 'negotiating stances' etc. But considering the lack of EU political process context given in the paper, the negotiating points that are established via the model do not seem to be embedded within the political reality of the EU system. Under some of the interpretations, some member states could theoretically justify an approach of aiming for no more emissions reductions under the 2030 framework. Given the policy, governance and political processes context, how could this be considered a feasible/fair negotiating stance? Remember the context of policy mixes where bargaining also happens across policy files; the context of aiming for climate neutrality by 2050...

- Returning to the point above about the interactions among the interpretations, I would really welcome far more critical engagement with the researchers' own findings, perhaps by embedding some of their thinking in, and by building on, the literature and politics/policy/governance/process discussions that I have mentioned here. If the authors were to engage critically in thinking on how the various interpretations may interact with each other (e.g. responsibility & renewable expansion with capability), I think the discussion would be far richer. Further critique on the choices the authors make in the construction of their approach and interpretations would also be welcome: what if other considerations were taken into account? The limited critical reflection of the approach presented is also evident from the abstract (lines 16-18), where the authors write that they 'demonstrate the applicability of our approach' without referencing critical engagement with the proposed approach.

Finally, some small points for correction/clarification:

- It is best to refer consistently to the 'European Green Deal' and not the 'Green Deal' as there are many different green deals around the world (line 7).

- What do the authors mean by referring to the approach as 'pure'? (line 51)

- Remove time references such as 'current', 'ongoing' (e.g. in Table 1) and replace with specific time phases (e.g. in 2021).

References

Boasson, E. L., & Wettestad, J. (2013). *EU climate policy: Industry, policy innovation and external environment*. Ashgate.

Capoccia, G. (2016). Critical Junctures. In O. Fioretos, T. G. Falletti, & A. Sheingate (Eds.), *The Oxford Handbook of Historical Institutionalism* (pp. 89–106). Oxford University Press.

Dupont, C. (2016). *Climate Policy Integration into EU Energy Policy: Progress and Prospects*. Routledge.

Gheuens, J., & Oberthür, S. (2021). EU Climate and Energy Policy: How Myopic Is It? *Politics and Governance*, 9(3), 337–347. <https://doi.org/10.17645/pag.v9i3.4320>

Gupta, J. (2010). A history of international climate change policy. *Wiley Interdisciplinary Reviews: Climate Change*, 1(5), 636–653.

Hall, P. A., & Taylor, R. C. R. (1996). Political science and the three new institutionalisms. *Political Studies*, 44(5), 936–957.

Jones, E., Kelemen, R. D., & Meunier, S. (2016). Failing Forward? The Euro Crisis and the Incomplete Nature of European Integration. *Comparative Political Studies*, 49(7), 1010–1034. <https://doi.org/10.1177/0010414015617966>

Jordan, A., & Matt, E. (2014). Designing policies that intentionally stick: Policy feedback in a changing climate. *Policy Sciences*, 47(3). <https://doi.org/10.1007/s11077-014-9201-x>

Mavrot, C., Hadorn, S., & Sager, F. (2018). Mapping the mix: Linking instruments, settings and target groups in the study of policy mixes. *Research Policy*, 103614. <https://doi.org/10.1016/j.respol.2018.06.012>

Oberthür, S. (1999). *The Kyoto Protocol. International Climate Policy for the 21st Century*. Springer.

Peters, B. G., Pierre, J., & King, D. S. (2005). The politics of path dependency: Political conflict in historical institutionalism. *Journal of Politics*, 67(4), 1275–1300. <https://doi.org/10.1111/j.1468-2508.2005.00360.x>

Rhinard, M. (2019). The Crisisification of Policy-making in the European Union. *JCMS: Journal of Common Market Studies*, 57(3), 616–633. <https://doi.org/10.1111/jcms.12838>

Schmidt, V. A. (2010). Taking ideas and discourse seriously: Explaining change through discursive institutionalism as the fourth 'new institutionalism'. *European Political Science Review*, 2(1), 1–25. <https://doi.org/10.1017/S175577390999021X>

Siddi, M. (2021). Coping With Turbulence: EU Negotiations on the 2030 and 2050 Climate Targets. *Politics and Governance*, 9(3), 327–336. <https://doi.org/10.17645/pag.v9i3.4267>

Zaun, N. (2016). Why EU asylum standards exceed the lowest common denominator: The role of regulatory expertise in EU decision-making. *Journal of European Public Policy*, 23(1), 136–154. <https://doi.org/10.1080/13501763.2015.1039565>

Reviewer #3 (Remarks to the Author):

I will caveat my comments from the start with: I am from a different discipline and am reading this largely to identify issues regarding logic, clarity, usefulness etc for policy/law/practice. I have gone nowhere near the maths or the methods! If my points are mistaken due to misinterpreting material, then apologies.

The content is interesting, valuable (theoretically and practically) and topical. It builds on three equity principles highlighted by the IPCC and could be very useful. The identification of the 10 components and their application, as well as subsequent analysis provides insights into alternative mechanisms in calculating reduction targets for emissions and possibly identifies foundations for future negotiations (this aspect needs further development/clarification). The piece can contribute to important on-going academic and policy debates and the proposed tool will be very interesting to see also. If the tool ends up being included in the eventual publication, it would be important to ensure that it is more broadly accessible (without having to pay journal fees), i.e. perhaps make it available on an institutional repository or similar (checking with funders and employing institutions first regarding ownership and the like) and include the link in the publication? (In case this is not what is already intended)

However, improvements could be made in particular regarding clarity and coherency – whether regarding readability, justifications, development of arguments or otherwise.

- Clarity: overall, the clarity could be improved across the piece. This relates to both individual sentences and paragraphs, but also the flow of the piece. In particular, if the abstract, introduction and conclusion were improved, this would help the overall piece considerably. However, the same need for clarity applies across the piece, e.g. later in discussing the application of the principles or negotiation convergence points (e.g. lines 278-283).

There is also the question to what extent you wish the content to be accessible to individuals from different disciplines. It took a few reads to understand some elements and I am not used to figures and graphs in my discipline, but they were manageable for the main part. Figure 3 could be clearer I think (not my field though) and perhaps continuing to use R1, R2 etc would be useful for consistency?

- Purpose: what precisely is the purpose of the paper? Is it theoretical or practical or both? Is it for the EU/Commission/Member States or globally or both/all? What is the end goal?
- Role of the principles (related to purpose of paper): how do you see these being used in the end?

You mention (briefly/as an aside) some limitations, but still support their use. So who is to use these, how/to what end will/could/ought they apply them (e.g. is there an overarching aim for those applying them), what limitations arise and how can these be addressed... etc? Or does any of this matter?

o To build on this: what's to stop individual Member States saying use e.g. R2, C1 and E3 for us please as that involves the least effort/change by them? Would you use different combinations for each Member State? If somehow you apply the 3 principles and you get a perfect balance that is acceptable to everyone (although nobody is happy with a compromise?) and will meet the overall EU targets on the button, what happens if one Member State then doesn't meet their obligations?

o The paper doesn't need to address everything, but it needs to be clear as to what it is seeking to address, the limits it is imposing on itself (write a follow-up paper if needed!) and the caveats that need to be taken into consideration.

- Justifications: these are frequently lacking and would be of assistance. Including: why this paper, why this focus, why these principles, why these facets and not others... this applies across the paper.

- Proofread: There are some slips/typos across the piece and a careful proofread will be required at the end. E.g. Pg2, line 22, 'need' not 'needs'. 'Broken down to'? 'Broken up and redistributed across' perhaps? The slips also affect the clarity at times.

Some more specific points:

- Why is C1 meant to mirror the EU approach?

- Are there other facets for each of the three principles? Why discount these? E.g. could you just say for equality: each State should reduce their outputs by X%? Or by size of the country? Similarly, for responsibility, why not go back longer and look at the harms done (rather than benefits garnered) irrespective of the timing of scientific reports?

- How different is E2/E3 from R3?

- Are there not concerns that some approaches could hinder overall efforts? Eg. R4 rewards past good behaviour by allowing increased levels of promotion? (I know this is reflected in several approaches, but it's more blatant here).

- Re basic needs –why multidimensional poverty? Why not simply needs or heating needs? (not objecting to this, but wanting the justifications)

- The different number of components for each principle is interesting – e.g. if seeking to weight R, E and C equally and using all components, does that lessen the significance of R1-5 for instance as there are 5 compared to 2 C and 3 E.

- What is a cathete? After some googling and consulting with some mathematicians, I think you mean cathetus (cathète in French)? But the mathematicians I spoke with don't use the term.

- Does the following point just hold for Germany or for every country? Either way, why? 'While increasing the weight of equality (i.e., moving down along the right hand cathete) reduces emission reduction targets – the reason being that an increase in the weighting of this dimension necessarily decreases the weight of the other two dimensions, each of which imply stronger reduction target increases.'

- It was meant to be possible to combine multiple or all factors – was this done? Is it useful? Rather than having numerous triangles for each country using just 3 components?

- What about using all R components and nothing from E or C for instance?

- The graphs supposedly enable the identification of country preferences – what do you mean (existing preferences or it allows countries to figure out what suits them best based on either criteria or the outcome?) and how does it do this?

- How does this facilitate 'identification of possible points of agreement'? (pg13, c. line 256) This is further developed on pg16, but some greater clarity is needed and cross-referencing from earlier on to the later section would be useful.

- What do you mean by 'these points indicate the weighted combination of equity interpretations which result in the lowest possible aggregate deviation of all EU countries from a given preference, e.g. the equity consideration which requires minimal mitigation effort.'? What is the 'given preference' here? Are you presuming that each Member State wants to make the least effort possible? How are you balancing this out between Member States? E.g. if countries with high emission levels (in absolute terms, not per capita) make very little effort, then other countries will have to make a disproportionately large effort.

- I'm not sure how the minimal effort is being identified for one route, but for the one linked to the ESD, the results are not simply unsurprising – they are surely expected? If one calculates the

minimal effort relative to the ESD and that is largely based on capabilities, the minimal effort is clearly also going to be largely based on capabilities. It's not exactly circular, but close enough. Regarding the other route: if this is linked to existing practices/measures, presumably many EU countries nowadays do what they feel they are able to? And EU policy doesn't appear overnight – MS would have been preparing and also influencing its development, as well as in light of international commitments more generally... Minimal effort seems to be something that is likely to be linked to capability...

- Should minimal effort be your focus?
- Is the logic: countries will want to make the minimal efforts possible; their minimal efforts that might still facilitate overall EU aims (this argument is weak, as seems to be based on existing balance and therefore quasi-pre-determined/biased – both re quantum and approach) can be identified using the 3 principles; specifically, those minimal efforts are typically based on X, Y and Z (e.g.C1); this common approach/understanding facilitates negotiations and agreements.....?
- Fairness is mentioned on p.17- is that the overall goal? How does it work with the principles?

Overall, I think there is valuable content, but the manuscript needs to be improved and re-worked considerably still. Once the issues are resolved, it has considerable potential to contribute to both academic debate and policy development.

Responses to reviewer comments for submitted article titled:

Sharing the effort of the European Green Deal among countries – *Nature Communications*

We would like to thank the editors and all three reviewers for the very thorough and thoughtful critique of our work, which we feel has helped us to greatly improve our submitted manuscript. Overall, the comments provided have enabled us to provide further clarity and a clear framing of purpose for our work and has led to an extension of the indicators and an improvement to enable a better understanding of our manuscript, and particularly in how we convey the conclusions and discussion of the potential political implications. As can be seen below, we feel we were able to address all comments proper and provide additional detail when requested (of note is that in the table below, reviewer comments are *in italics*, and responses are **blue colored text**, with direct citations from the revised manuscript **in green colored text**).

In addition, we supply a file of the manuscript where all changes to the original submission are marked (track change). The Supplementary Information consists of fully new content, with new sections 1-4, and with updated country ternary charts (now section 5) in terms of both style and extension via further interpretations.

Given that all reviewers were supportive of our work and the new insights it supplies, provided we make substantial changes to clarify the issues raised, we hope that the editors will find the revised manuscript acceptable for publication in *Nature Communications*.

Sincerely,

The Authors

REVIEWER COMMENTS

Reviewer #1 (Remarks to the Author):

The article 'Sharing the effort of the European Green Deal among countries' proposes a new analytical framework to assess and distribute emission reduction commitments among several actors based on three justice principles, namely capability, equality and responsibility.

In general, the proposed framework (though I am not an expert on the applied methods) is interesting and largely convincing. However, I do have some suggestions for improvement.

- | | |
|-----|--|
| 1.1 | 1. My main point is that the conclusion and discussion of the political implications of the framework is a bit short and under-complex. In the conclusions, the authors argue that the framework could help to identify acceptable negotiation outcomes. While this might be true, the political negotiations within in the EU are not only driven by considerations of fairness/economics. Additionally, they are not confined to the climate realm. Meaning that the commitments of individual states are influenced by side-deals, historical path dependencies, power struggles, different cultures, technological capabilities, domestic politics etc. Therefore, even though the proposed framework might be quite straightforward in purely rationalist/economic terms, it remains to be seen whether it really would make a difference in real-life political negotiations. For instance, if some countries would see a chance to lower their ambition based on some of the proposed mechanisms (Bulgaria, Romania), others would certainly object to using these particular mechanisms. Equally, if some countries would have to increase their commitments even though they already have the highest commitments, they would most likely try to block frameworks that use these mechanisms. Eventually, opening up the currently agreed upon mechanisms (mostly |
|-----|--|

capabilities), could lead to cherry picking of mechanisms by the countries and seriously complicate or derail the already fragile EU internal climate negotiations.

In my opinion, the article should at least discuss these limitations in the conclusions. It could also be interesting to discuss some of these limitations in more detail in relation to an exemplary country (e.g. the authors could add that to the section on Germany). It would be even more interesting to compare the opportunities and possible hindering circumstances in two different states, preferably one where a different allocation between the mechanisms would lead to an increase in commitments and one where it would allow for a reduction.

Thanks for pointing out the relevance of the framework embedding and the political negotiation process more explicitly. We have expanded the discussion of the embedding of our analysis, mainly both in the introduction and in the conclusions (also taking advantage of the suggestions of reviewer#2, point 2.1.1).

The section in the introduction now reads:

Building upon the categorization of equity principles used in the effort sharing literature⁶⁻⁹ we present an approach that can contribute to a transparent decision process when partitioning an overall emissions target among subsidiary entities by aligning disparate views and defining an allocation space where different parties could agree to an equitable compromise. For negotiation processes heavily influenced by a range of factors, such as historical path dependencies¹⁰, side deals, power struggles, different cultures, or domestic politics, such transparency in terms of equity may be a significant factor contributing to success, given the increasing weight of the equity dimension in such negotiations¹¹.

The conclusion now also includes:

We note that the results of our approach can be employed to inform parties in ongoing political processes but do not serve as a projected end-point for negotiations, due to a number of factors. Our aim is to provide a framework for distributing future emissions budgets based on well-established equity principles, but it must be emphasized that contextual factors, the choice of interpretations and their implementation can lead to differing outcomes. Numerous potential proxies could be suggested for a given interpretation (see Supplementary Information section 2 for discussion on possible governance indicators). Application of increasing renewable shares, as either a responsibility or capability interpretation, is an apt example; valid arguments can be made to place it under either equity consideration, but arguments against are also relevant. Implementation of an RES-based capability interpretation is problematic as it can be seen to reward countries for a lack of past effort without encouraging lagging countries to act by increasing their reductions. Also, past performance regarding RES is not a reliable indicator of future performance, as political or economic conditions may lead to changing capability. For a further discussion on the ambiguity of renewables as an equity interpretation, see the Supplementary Information, section 4.

[...]

Given the above issues, our approach provides a framework for discussing equity-compatible 2030 targets, but it is up to negotiators to not only choose the weighting amongst the three equity dimensions (and their respective interpretations) but also to decide how strongly an interpretation influences emissions reduction. On the latter we have consistently determined allocations based on percentage deviations of interpretations from an EU

	average, where applicable (i.e., for all interpretations not based on budget approaches, directly translating to emission reductions by country; see Methods for further details). While we do not analyze the actual political processes, the related governance literature has informed our analysis. First, it identified the particular relevance of path dependency, i.e., the notion that policy decisions once made within a certain frame tend to stick to that frame^{10,50}, which guides our choice of past EU effort sharing regulation in one approach to derive negotiation convergence points. Second, the governance literature identifies the relevance of timely transparent information. In particular, the 2019/2021 EU decision of implementing to remain within a Paris compatible carbon budget in combination with the Corona-aftermath and “building back better” may represent a critical juncture in institutional development, at which “decisions of important actors are causally decisive for the selection of one path of institutional development over other possible paths”⁵¹, and conflict over ideas has been identified as important for institutional change⁵². The approach we present is intended to contribute to resolving conflicts over equity perspectives to allow for institutional development. The results of our framework make transparent how different choices of equity interpretations can translate into different country contributions. The range of transparent equity considerations made explicit here allows for appreciating the positions of other countries, as well as for a common understanding of the range of outcomes, and thus can contribute to successful negotiations. The discussion/comparison of different states we have introduced respectively expanded at a different location: given that the extensions in response to the many highly relevant issues by reviewers brought us over the word limit, we had to cut back otherwise, and did so also at the exemplary introduction of the ternary chart using Germany to now only cover two panels in Figure 3, still fulfilling the purpose of introducing the concept here. We thus expanded on the discussion across countries later, specifically in the context of the overall results of Figure 4 (in the section immediately following the one with Figure 3, i.e. in section “Possible negotiation convergence points”), as well as expanded on it in the conclusions.
1.2	2. The authors base their three effort sharing mechanisms on three well-known, quite broad justice principles. However, they then narrow them down to specific variants, for instance the capabilities mechanism to the ability to pay principle and then to the GDP of individual actors. One could argue that there are different operationalisations of the capability principle e.g. by Schlosberg and others, that do not necessarily look at the GDP but rather on specific capabilities of countries to contribute to climate abatement and climate justice. These do not necessarily have to be linked to the GDP but could consist in organizational, diplomatic, or technological capabilities. This again boils down, to the above-mentioned tendency of the article to treat countries as rationalist like-units, which could at least be discussed more openly. We agree and have expanded in particular for a broader set of capability indicators. First, directly on the issues indicated here, we have included a further interpretation, (C3-Governance), in our analysis, based on the World Bank’s Worldwide Governance Indicators, specifically estimating government effectiveness. This index takes into account a number of aspects of e.g. quality of public services, policy formulation, infrastructure and human capital. A description of this alternative interpretation can be found in Table 1 and in the description of equity principles; the relevant section of the latter now reads:

	While interpretations focusing on macroeconomic indicators do address the ability to pay in a very literal sense, the ability of an actor can be argued to extend beyond GDP. Issues of governance, and the ability for an actor to effect changes, rely also on institutional effectiveness, human capital, bureaucratic quality, and other aspects not taken explicitly represented by GDP. To incorporate these, indicators of government effectiveness (such as [26]) can be used to incorporate differences in perceived quality of public and civil services or quality of policy formulation to emission reduction needs (C3-Governance, see Table 1 and SI section 2 for details). The new interpretation is included in our analysis, and thus results can be found in Figures 2, 3, 4 as well as discussion in the results and conclusions sections. To select this indicator, we first tested for a broader range of institutional capacity indicators, as reported now in detail the SI, giving the respective implication on emission targets for each of them when implemented within our algorithm. We then selected the above indicator for it being most different in its implication to GDP/capita, as described in the SI. Finally, and this also in response to reviewer#2 comments, we added a fourth capability indicator, renewables extension, also in this alternative interpretation: Alternatively, the capability to reduce emissions in the future could be reflected in recent achievements in building up renewable energy capacity. The expansion of renewables could indicate increased ability to reduce emissions relative to other countries, whether due to circumstances such as advantageous natural resources or technological know-how resulting from early adoption and the consequent ability to efficiently expand the use of renewable energy sources (RES) in the future (first mover advantage). In that sense, RES expansion could be interpreted as an indicator of capability; when countries have succeeded in improving emission efficiency by means of implementing renewables, they could be considered more capable of further reducing emissions and, thus, should be allocated a smaller share of the remaining budget. We thus include an interpretation placing greater emissions reductions on countries most likely able to meet such demands, as represented by recent development of RES (C4- RES-cap). (see our response to comment 2.2.2 for further details).
1.3	3. The claim that states were only liable to know about the harmful consequences of GHGs from 1995 can of course be disputed, they already knew from the late 1980s or at least since the 1992 UNFCCC with a considerable level of certainty. We recognize that our claim that states were liable from 1995 onwards is debatable, and other years could be argued for on solid grounds, for reasons you provide and others. However, we investigated the implications of changing the historical year at which accounting starts, and find that it is the imposition of such a historical consideration, rather than the precise year, which makes the most difference. In the Supplementary Information, in a new section 3, we now provide a figure which shows the resulting reductions required of countries if the starting year for accounting would shift between 1990 and 2000 (we do not project further back in history due to issues with data). The countries shown in the figure are the only ones which would experience any change over the time period. This is for two reasons; for many countries, a historical consideration would require them to have negative emissions by 2030. We do not consider this feasible, and thus

	limit the reductions to 100% of 2005 emissions. Conversely, countries which have emitted relatively little in the past would be allocated emissions far beyond their 2005 levels. As this is not in line with broader climate goals to 2050, and would then lead to difficulties in shifting from a decade of rising emissions to stringent 2050 net zero targets, we limit countries to have no positive change in emissions compared to 2005. For both groups of countries shifting the starting year of accounting for historical emissions around 1995 (up to 5 years earlier or later) does not change the respective restriction to be relevant, and thus does not change the resulting 2030 emission reduction target. For the remaining countries, only a few (e.g. France, Greece, Czechia, Cyprus and to a lesser extent Germany) would see substantial (~10% or higher) changes in their emission allowances compared to a 1995 base year. The others only have minor fluctuations, and, as mentioned, the remaining 20 EU countries do not experience a change at all, for the reasons discussed above. To make this clear to the reader, we have added a section in the main text: First, when introducing the year of reference in the “equity principles” section: One could cite other dates for good reasons, e.g. the 1992 ratification of the UNFCCC at the Earth Summit or the late 1980’s establishment of the IPCC, which we assess in more detail in the Conclusions section and in the Supplementary Information, section 3. Second, a discussion in the conclusions section: Beyond choice of an interpretation, its application can have varying effects on outcomes. The most obvious example is the question of when to start taking historical emissions into account; we choose 1995 based on publication of the IPCC’s Second Assessment Report. However, valid arguments can be made that other, particular earlier, years should be chosen. We find that while that is the case, shifting the year has less of an effect on outcomes compared to the initial choice to consider countries’ historical responsibility. Overall emissions reductions for countries are mostly unaffected or would see only minimal changes (see Supplementary Information, section 3 and Figure SI.2 for further detail). For the specific case of the EU, both the historical emissions and inherited benefits (R1 or R2) interpretations acknowledge the specific context at the time, with Eastern European countries having been comparatively emission intensive at lower efficiency up to 1990 with emissions plummeting thereafter. Historic emissions consider aggregate emissions over the whole period back to 1995, but not including the high-emissions period up to 1990. Benefits received are derived from capital stock available in 1995 (i.e., after economic restructuring), and are evaluated using a recent average EU emission intensity (and not the historical – and more emission intensive – levels from the years before 1990). as well as have included the more detailed discussion and respective figure in the Supplementary Information (section 3).
1.4	4. Some of the figures are quite hard to read, especially if one reads the paper in B/W. Thank you for pointing this out; we have altered our figures to be B/W printing friendly (including changing colors and for Figure 2 a switch to fully B/W) and have tried to improve clarity more generally.

Reviewer #2 (Remarks to the Author):

I was very pleased to have the opportunity to review this paper. It covers a timely topic and presents a considered framework for assessing how to think about sharing the effort in meeting the 2030 greenhouse gas emissions reduction target in the EU. I was therefore enthusiastic when I received the invitation to review.

I would first like to commend the authors on their paper. Their efforts to present an approach show the complexity in considering questions such as ‘fair share’ or ‘effort sharing’ and the distribution of such efforts in theory and in practice. I am not an expert in previous studies on effort sharing distribution calculations, and I learned a lot in reading this paper.

While being enthusiastic about the topic and approach of the paper, I was, however, left dissatisfied when I came to the final page of the paper. Perhaps since I am rather a researcher of climate policy and governance in the EU more broadly, I felt some of the weaknesses of the approach and the gaps in the discussion flow from the too-tight connection only to effort sharing and a focus on the 2030 timeframe. From this sense of dissatisfaction as a reader, I offer my feedback and comments in the hope that the authors will find them useful or insightful. At the same time, I acknowledge that some of my comments may at times seem to the authors to be outside the scope of what they are trying to achieve in the paper, but I would offer that as my first comment: perhaps it would be worth reflecting on the scope – what is the objective of the paper and can it be achieved within the scope set by the authors?

Thanks indeed for the so comprehensive and detailed comments as listed below, also pointing out and opening access to related strands of literature! We respond in detail below, but most fundamentally clarified the objective of our paper now explicitly in the introduction. The introduction now states

In political processes, quite often this effort sharing of a common target is hampered by diverging viewpoints and perspectives on how to determine a fair share. Building upon the categorization of equity principles used in the effort sharing literature⁶⁻⁹ we present an approach that can contribute to a transparent decision process when partitioning an overall emissions target among subsidiary entities by aligning disparate views and defining an allocation space where different parties could agree to an equitable compromise. For negotiation processes heavily influenced by a range of factors, such as historical path dependencies¹⁰, side deals, power struggles, different cultures, or domestic politics, such transparency in terms of equity may be a significant factor contributing to success, given the increasing weight of the equity dimension in such negotiations¹¹.

[...]

However, our proposed framework allows for systematically combining different interpretations of three major equity considerations – capability, equality, and responsibility – and allows us to explore compromises, where country targets are the result of a combination of different allocation schemes, thus weighting different allocation principles to find a common solution.

I list the comments in no particular order below. In summary, the comments refer to: 1) framing of the paper; 2) choices in the approach and interpretations; 3) discussion and conclusions.

2.1	1) Framing of the arguments, assumptions, objectives and discussions in the paper Here, I highlight some of questions that came to me as I read the paper’s introduction, discussion and conclusions, and how these questions also connected to the approach put forward in the paper.
-----	---

2.1.1	- In both the opening of the article and in the concluding remarks, the authors refer to the EU process of dividing up the effort to reduce GHG emissions in sectors not covered by the ETS as ‘ad hoc’ and the result of ‘political processes’. Indeed, I agree that the outcomes are the result of political processes, but I would not necessarily consider these outcomes to be ‘ad hoc’. At the same time, I think the authors themselves (given their results) would agree that these are outcomes based within a certain proposed framing that draws upon a logic of capability, and not from an ad hoc political process. Furthermore, if the authors were to step outside their scientific circle, they could learn quite something about how EU political processes are themselves often less than ad hoc. In EU governance studies, we find research highlighting the importance of path dependency: the notion that policy decisions once made within a certain frame tend to stick to that frame. So, if a capability focus is the basis for effort sharing (or even earlier, burden sharing) decisions in the past, then it is likely that it would remain in the present/future unless major events or ideas (critical junctures, crises, e.g.) intervene to shift the policy development framework onto another path. This is best laid down theoretically by researchers on historical institutionalism, and also perhaps discursive institutionalism (Capocchia, 2016; Hall & Taylor, 1996; Peters et al., 2005; Schmidt, 2010). We fully agree that especially within the EU context our earlier wording of an “ad hoc” decision was misleading without specifying closer, and fully revised and expanded our discussion of the embedding into and potential benefit for the context of the political process. In that respect we in particular revised the introduction. We now also discuss the link to the governance literature in conclusions. While we do not directly contribute to that literature, we can build upon some of its results, which we indicate especially in the conclusions. Two of the paragraphs in the conclusions section now read: While we do not analyze the actual political processes, the related governance literature has informed our analysis. First, it identified the particular relevance of path dependency, i.e., the notion that policy decisions once made within a certain frame tend to stick to that frame^{10,50}, which guides our choice of past EU effort sharing regulation in one approach to derive negotiation convergence points. Second, the governance literature identifies the relevance of timely transparent information. In particular, the 2019/2021 EU decision of implementing to remain within a Paris compatible carbon budget in combination with the Corona-aftermath and “building back better” may represent a critical juncture in institutional development, at which “decisions of important actors are causally decisive for the selection of one path of institutional development over other possible paths”⁵¹, and conflict over ideas has been identified as important for institutional change⁵². The approach we present is intended to contribute to resolving conflicts over equity perspectives to allow for institutional development. The results of our framework make transparent how different choices of equity interpretations can translate into different country contributions. The range of transparent equity considerations made explicit here allows for appreciating the positions of other countries, as well as for a common understanding of the range of outcomes, and thus can contribute to successful negotiations.
2.1.2	- Furthermore, continuing the discussion of EU political processes, there is an implicit assumption in the approach outlined in the paper that member state preferences would be to align their negotiation position to achieve minimal effort (from line 276 and in figure 4). This implies an assumption that decision-making in the EU tends towards what scholars would call a ‘lowest common denominator’ outcome. Certainly, in crisis situations there seems to be evidence that the EU advances through such lowest common denominator

	decisions, but that is not necessarily a rule. In fact, much research on the nature of EU political and policymaking processes (when not in crisis response decision-making mode) has shown that these processes tend to lead to outcomes that are higher than the lowest common denominator and tend towards an advancement of what can be called the ‘community interest’ (Jones et al., 2016; Rhinard, 2019; Zaun, 2016). Explanations for why the EU has regularly adopted policies beyond the lowest common denominator can again be found in the institutional set-up and institutionalist perspectives described above, but not only. The role of expert groups, the deliberative nature of the decision-making procedures (which tend to be set aside or at least less central in crisis situations and hence why crisis decision-making may display more lowest common denominator outcomes than normal procedures), notions of policy interconnections or spillovers, and the creeping competence of the EU level (more assertiveness and power to the European Parliament and/or European Commission) through various treaty amendments and functional policy advancement can play a role. From the perspective of the European Green Deal, I also would like to highlight the importance of the EU’s policy mix approach to reaching an overarching objective. By agreeing on an overarching objective, and by negotiating on a package of measures that together aim for that overarching goal, policy makers are both constrained and free to negotiate their preferences. First, they are constrained because they will have to contribute to the overarching goal, to which they also have agreed. Second, they are constrained and freed by the possibility to choose their battles and engage in bargaining across policy files (Boasson & Wettestad, 2013; Dupont, 2016; Jordan & Matt, 2014; Mavrot et al., 2018). We also agree that our previous version of the paper implicitly gave substantial weight to the idea that country preferences are based on minimizing effort. This was an error on our part, and we have changed the way we discuss our derivation of negotiation points. In using and comparing (a) the least deviation from the upper bound of equitable-compatible emissions, and (b) least deviation from the 2018 ESR, we want to reflect the relevance of aspects of path dependency, due to likely weighing on planning and policymaking processes, and can show how relevant these have been in the past (a new discussion located just before Figure 4). Imposition of strict, unforeseen new targets would be more likely to be met by resistance than solutions which required less deviation. In particular, and now made explicit in the manuscript, as the 2018 ESR has been a target for a number of years, it is reasonable to think that countries have been planning with these goals in mind, and would be more likely to be on track for reductions closest to the original ESR, rather than being vastly different. To reflect the above, we no longer refer to our negotiation points as ‘minimal effort,’ but now instead refer to them as “closest to upper bound of equity-compatible emissions” and “a minimal change from the 2018 ESR”. Numerical results changed accordingly (mainly Figure 4, but also the respective negotiation points in Figure 3 and the SI ternary charts). The respective changes have been made throughout the text – predominately in the Results (specifically, Possible negotiation convergence points) and Conclusions sections and methods, as well as in captions for Figure 3 and 4 (and in the legend of Figure 4).
2.1.3	- A last point on context: it seems important to consider the context for the effort sharing agreement towards 2030 within the overarching 2050 climate neutrality goal. The 2030 goals serve as stepping-stones or as intermediate goals towards the 2050 goal of achieving climate neutrality in the EU. Further, the EU accepts the scientific advice of the

IPCC that the transition to climate neutrality needs to be under way within the decade to 2030. Given this overall goal and the broader context, the effort sharing negotiations also should be seen within this. There is therefore a temporal consideration to think of in the effort sharing discussion: if we think of the 2030 goal as a stepping stone towards the 2050 goal, it only makes sense that all member states make some effort in this decade as part of the process towards 2050. I think this broader context of the longer-term goal should be considered also. Would it not lead to some adjustment in the interpretations (e.g. a baseline of 'fairness' would be that all contribute at least something to emissions reductions in areas outside ETS to ensure they start on the path toward climate neutrality in a timely enough manner)? The EU has struggled at implementing long-term thinking into its policy developments in the past, but there have been some improvements in this regard (Gheuens & Oberthür, 2021; Siddi, 2021).

To address this crucial issue specifically for the EU in the discussion of our results, we expanded the conclusions, now including:

Approaches such as presented here also need to be embedded in larger policymaking contexts; the communication of the EU Green Deal emphasizes that the combination of the climate neutrality goal by 2050 and ambitious 2030 climate targets together act as a crucial framework to provide long-term certainty and predictability for investments⁴⁹. Considering the 2050 perspective, we note that for some countries a small subset of negotiation points result in zero emissions reductions compared to 2005 levels, which might imply allowances for increasing emissions until 2030. While most of the countries this refers to had higher emissions in 2019 than in 2005, and thus would still be required to reduce emissions, it does not hold true for three countries (Romania, Croatia and Greece). Here, a minority of weighting combinations might allow rising emissions. Such a development would need to be considered in the context of the EU net zero emission target by 2050, which – given the limited potential of negative emissions – essentially translates to a close-to-net zero target for every country. Thus, equity considerations may prohibit rising emissions up to 2030, particularly if this would imply an increase in future stranded assets, and would need focused deliberation.

More generally, we have tried to strike a balance between contextualizing our approach and results for the current and future EU climate goals, while on the other hand proposing a flexible method which can be applied in other contexts than the EU, an approach based on combining well-recognized equity considerations via a range of possible interpretations.

In this sense, we aimed to avoid introduction of arbitrary or ad-hoc constraints into our approach and results. We see this approach not as perfectly incorporating all considerations or contextual factors surrounding EU climate target discussions, but rather as forming a starting point for discussion, based on well-established fairness principles.

In this sense, results such as you refer to, where member states may not need to make an effort in this decade, may not be acceptable in the broader context, but such outliers are insights and could ideally be taken into account in negotiations, which will doubtless include other criteria we cannot include here.

However, we do introduce two constraints to our algorithm in order to capture some aspects of the broader context you suggest, which is now made better explicit in the methods section. While not tailored specifically to the EU context, it does touch upon the issues you mention. First, we do not allow for countries to have higher emissions in 2030,

	relative to 2005. Second, we do not allow for negative emissions in 2030, as it is not foreseen as feasible within a 10 year period to have such technology working at scale.
2.1.4	- I also wonder about the overall objective of the paper. It doesn't seem fully clear, yet. Is the purpose to figure out if the approach taken in the EU (focus on capability) is 'correct' or 'fair'? Is the purpose to check if an alternative approach would be 'fairer'? Is the purpose to reveal potential negotiating stances based on alternative approaches and prepare for that? And (how) does this contribute rather to scientific knowledge, and not only to the policy context at the moment? All of these may seem valid, but it is not really clear yet what the precise objective here is. Your observation (as well as a similarly directed one of reviewer#3, comment 3.2) made us aware that the overall objective was indeed not clearly stated before. We now revised the introduction to clarify and eliminate this earlier shortcoming, and expanded the conclusions to indicate how the new type of scientific results can be informative in the policy process. We copied the respective segment of the expanded introduction in response to the very introductory comment above (at the beginning of this section of reviewer #2. The conclusions now also include: We note that the results of our approach can be employed to inform parties in ongoing political processes but do not serve as a projected end-point for negotiations, due to a number of factors. Our aim is to provide a framework for distributing future emissions budgets based on well-established equity principles, but it must be emphasized that contextual factors, the choice of interpretations and their implementation can lead to differing outcomes. Numerous potential proxies could be suggested for a given interpretation (see Supplementary Information section 2 for discussion on possible governance indicators). Application of increasing renewable shares, as either a responsibility or capability interpretation, is an apt example; valid arguments can be made to place it under either equity consideration, but arguments against are also relevant. Implementation of an RES-based capability interpretation is problematic as it can be seen to reward countries for a lack of past effort without encouraging lagging countries to act by increasing their reductions. Also, past performance regarding RES is not a reliable indicator of future performance, as political or economic conditions may lead to changing capability. For a further discussion on the ambiguity of renewables as an equity interpretation, see the Supplementary Information, section 4. [...] The results of our framework make transparent how different choices of equity interpretations can translate into different country contributions. The range of transparent equity considerations made explicit here allows for appreciating the positions of other countries, as well as for a common understanding of the range of outcomes, and thus can contribute to successful negotiations. For regions without previous effort sharing agreements to refer to, as available for the EU, such transparent and commonly available exploration of the negotiation space is likely to be an even more important ingredient in the process to agree on sharing among subsidiary entities. We copied further central segments of the expanded conclusions now addressing this issue in our response to comment 2.1.1 above.

2.2	2) Choices in the approach and interpretations Now, I do of course understand that a discussion of the political and policy processes of the EU is outside the scope of this paper. At the same time, I think it is important that the authors realise that some of the messages or assumptions that seem to exist under their choices should be better justified for the reader. In particular, when it comes to the choice of interpretations of the three main considerations – capability, responsibility and equality – the fact that the authors do not place these within the context of the EU decision-making processes or the broader context of the history of EU effort sharing decisions within a policy package approach raises some questions. Correct, we had intended not to be too EU specific – but your point is of course highly relevant, and we thus included a new section on the history of EU effort sharing (right after the introduction, with further details in the SI), which is both an interesting case of a sharing implementation development, and of substantial relevance for the readers of our analysis to give them the adequate background. First a general note: the authors mention that their approach focuses on different equity dimensions, but they write about equality rather than equity throughout their methods, results etc. Equity and equality are not the same, perhaps this warrants some reflection. Thank you for this point. We agree. Equality and equity/ justice are not the same: an equal distribution is not necessarily a just or equitable distribution. In line with IPCC AR 5 WG III, Ch. 3 (Kolstad et al., 2014), and the answer given there to “FAQ 3.2 Do the terms justice, fairness and equity mean the same thing?” we make explicit the overarching character of equity or justice (and interpretations reflecting considerations of equality, capability and responsibility) already in the introduction. In particular we revised the first and added the second sentence in the following passage of the introduction: “However, our proposed framework allows for systematically combining different interpretations of equity or justice on the basis of three major principles – capability, equality, and responsibility – and allows us to explore situations where country targets are the result of a weighted combination of interpretations of those principles, where member state reductions add up to the overall EU reduction target. In doing so we distinguish interpretations of equity or justice that reflect considerations of equality, capability and responsibility.” We further differentiate these terms in the section “Equity principles” and in the methods section. We believe to be explicit in the use of these terms and consistent throughout the text: We have rewritten the introduction to our Methods section, and in places where equity, or interpretations of equality, responsibility, and capability are discussed, have been clear in differentiating our use of these terms. On the choices, let me elaborate with some comments on the three considerations.
2.2.1	- Responsibility: Because political and policy processes in the EU are embedded within a historical and institutional context, some of the interpretations presented under the ‘responsibility’ consideration raised my eyebrows. First, I do wonder at the wisdom of choosing 1995 as a starting point for which countries should be liable for their emissions because of their knowledge at this time. In 1995, the EU enlarged from 12 to 15 members with Austria, Finland and Sweden joining that year. The EU was therefore a western European bloc at this time, and in the international climate negotiations, those countries

	that had emerged from the USSR had a special status as ‘economies in transition’. The GHG emissions in these countries plummeted in the early 1990s, because of economic crisis. The understanding of common but differentiated responsibilities and capabilities at this time (up to and including the agreement on the Kyoto Protocol in 1997 and its implementation to 2008-2012) was that those global industrialised nations that were deemed to have most benefited from and contributed most to historical GHG emissions were most responsible for the problem and the solution. In 1995, economies in transition and developing countries were considered justified in their continued GHG emissions because of pressing economic development needs – the industrialised countries were supposed to take on the bulk of the effort. So, how can a 1995 start date for liability apply to Romania at the same time as to the Netherlands? This was not the context of the time. Of course, industrialised countries failed, even if they did take on some effort. Thanks for pointing out this specific EU context, that we took advantage of and now also discuss in the conclusions section, now including: Beyond choice of an interpretation, its application can have varying effects on outcomes. The most obvious example is the question of when to start taking historical emissions into account; we choose 1995 based on publication of the IPCC’s Second Assessment Report. However, valid arguments can be made that other, particular earlier, years should be chosen. We find that while that is the case, shifting the year has less of an effect on outcomes compared to the initial choice to consider countries’ historical responsibility. Overall emissions reductions for countries are mostly unaffected or would see only minimal changes (see Supplementary Information, section 3 and Figure SI.2 for further detail). For the specific case of the EU, both the historical emissions and inherited benefits (R1 or R2) interpretations acknowledge the specific context at the time, with Eastern European countries having been comparatively emission intensive at lower efficiency up to 1990 with emissions plummeting thereafter. Historic emissions consider aggregate emissions over the whole period back to 1995, but not including the high-emissions period up to 1990. Benefits received are derived from capital stock available in 1995 (i.e., after economic restructuring), and are evaluated using a recent average EU emission intensity (and not the historical – and more emission intensive – levels from the years before 1990). Our approach seeks to be generally applicable, also beyond the EU27 context, but this specific EU27 example allows to explicate the specific implications of the first two responsibility indicators of Western versus Eastern European countries in their historic context, and what renders these particular indicator choices to tend to acknowledge the specific context at the time well.
2.2.2	Second, and here is where the interpretation on renewables also seems somewhat strange to me, when member states made some effort to expand their renewables sector, in the authors’ approach, this leads to an interpretation that rewards them by adjusting their share of effort downwards. But industrialised nations generally got a head start in this expansion because of their same (failed) commitment to take action on climate change so that the economies in transition didn’t have to. You could just as easily frame renewables expansion as a benefit from transitioning away from GHG emissions, or as a first-mover advantage, rather than focusing on the benefits only of the GHG emissions themselves. Thank you for this important suggestion and critical remark. We now provide an analysis of renewables both in terms of responsibility and capability. In the Conclusions section we

	discuss the pros and cons of understanding renewables in terms of capability, also providing a reflection of the limits of what our approach aims at: We note that the results of our approach can be employed to inform parties in ongoing political processes but do not serve as a projected end-point for negotiations, due to a number of factors. Our aim is to provide a framework for distributing future emissions budgets based on well-established equity principles, but it must be emphasized that contextual factors, the choice of interpretations and their implementation can lead to differing outcomes. Numerous potential proxies could be suggested for a given interpretation (see Supplementary Information section 2 for discussion on possible governance indicators). Application of increasing renewable shares, as either a responsibility or capability interpretation, is an apt example; valid arguments can be made to place it under either equity consideration, but arguments against are also relevant. Implementation of an RES-based capability interpretation is problematic as it can be seen to reward countries for a lack of past effort without encouraging lagging countries to act by increasing their reductions. Also, past performance regarding RES is not a reliable indicator of future performance, as political or economic conditions may lead to changing capability. For a further discussion on the ambiguity of renewables as an equity interpretation, see the Supplementary Information, section 4. Within the SI (section 4) we now discuss the ambiguity in more detail, and also show how the different versions of interpreting renewables expansion differ in their impact on translating to an emission reduction target across countries.
2.2.3	Lastly on responsibility, I do not know how or even whether it is advisable to think about this, but responsibility for emissions today is calculated based on the geographic location where emissions are produced. However, there are good arguments for considering historical emissions based on where the emissions are consumed. There is still a division in the EU among member states in this regard. Even if you cannot or would not bring in this dimension when it comes to considering responsibility, I do think it warrants a sentence of two of explanation. Some references on the historical context for international climate governance (Gupta, 2010; Oberthür, 1999). Yes, indeed, switching to consumption-base emission accounting would change results for the one responsibility interpretation R1 (historic emissions), which we now mention in the context of interpreting Figure 2 and country classifications. Given the specific split of the EU 27 in which one dominates the other, a switch to consumption based accounting would not qualitatively change the overall results but would quantitatively lower commitments when following R1 for all countries for which consumption-based emissions are lower than consumption based emissions (as now listed explicitly in the manuscript) and raise them for all other countries. The new paragraph in the results section reads: Note, that in accordance with the history of global and EU emission negotiations all interpretations are based on production-based emission accounting. For historic responsibility (R1) quantitative results would differ, if based on the alternative consumption-based emission accounting, which allocates emissions of the full value chain to the country of final demand, irrespective of where the emissions physically have occurred during the production process. For the EU27 for all but four countries consumption-based emissions are higher – those four being Bulgaria, the Czech Republic, Denmark, and Poland^{47,48}. Thus, while a switch to consumption-based accounting would increase reduction commitments under R1 for the former countries, but decrease for the

	four latter ones, the classification of the three groups would not change.
2.2.4	- Equality: Next, let's talk about the equality consideration. I appreciated the different interpretations here, and I felt that the authors have done a good job of trying to operationalise these interpretations. I was struck in the paper by the comment (lines 74-75) that 'according to the equality principle everyone should be able to enjoy a level of wellbeing above the level required to secure basic needs', and yet the interpretation as presented then focuses on achieving basic needs (not above basic needs). In the methods section, I learned that this is understood to mean sufficient nutrition, housing and transportation to meet established standards of living (lines 453-454), which is seen as in line with the sufficiency principle: is this equivalent then to 'reaching the poverty line', which is considered the proxy (line 77)? Because this still doesn't seem to meet the 'above' basic needs level, and so somehow misses the 'equality principle' benchmark. Could you explain better why you make this choice? What would it change to really implement the equality principle along the above basic needs interpretation? We understand sufficientarianism as follows: According to sufficientarianism what matters with high priority is that all people enjoy at least a level of wellbeing at the level of sufficiency. We base our analysis on basic-needs sufficientarianism according to which people enjoy level of wellbeing at the level of sufficiency when they enjoy a level of wellbeing above or at the level required to secure basic needs. This connects to the equality principle when you understand the principle to treat people as equals: All people have an equal (and very strong) claim to reach at least a level of wellbeing at the level of sufficiency. If sufficiency is defined in terms of the secure fulfillment of basic needs, then one needs to provide an interpretation of basic needs, what counts as basic needs and how they can be fulfilled under the current conditions in the EU. For that we rely on an analysis of people reaching the poverty line. We have included a sentence in Table 1 explaining our use of a multidimensional poverty index. As fulfillment of basic needs cannot be directly measured, the focus on multidimensional poverty is to reflect the multitude of satisfiers people commonly rely on to meet their basic needs. In a first step, each member state is pre-allocated the emissions equivalent to energy use (at current emission intensities) required to meet basic needs. After this initial step, the remainder of the budget is distributed to states in an equal-per-capita manner, so that all Member States are assigned at least enough emissions to reach the basic needs threshold, and then move beyond them. [formatted bold only here for indicating the issue]
2.2.5	- Capability: Finally, on capability, I understand also the choices made here in the interpretations of capability as 'ability to pay'. At the same time, I do also see a missed opportunity for the authors to conceptualise, consider and perhaps even critique their own work based on the interactions among the three considerations. For example, the interpretation of responsibility that includes RES expansion as an opportunity to reward states by allowing them a lower reduction target also could be considered differently: as historical evidence of their technical capability to reduce emissions (perhaps because of technical know-how or their geographical, territorial positioning or climate that means they have a high potential for RES on their territory). Interactions among variables is not just combining them, but it is investigating how, whether, why some more have more influence in some circumstances, what sort of interaction exists, whether the interaction bolsters or negates other interpretations etc.

	Thank you for your criticism and suggestion. For the specific issue on RES, please see above our answer to 2.2.2. More generally, we have strongly expanded the discussion of and reflection on our approach, especially in the conclusions section, especially also now discussing the interactions among interpretations, including referring to a range of new sections in the Supplementary Information for those interested beyond the argument itself in more details, both of which seems indeed crucial for this article.
2.3	3) Discussion and conclusions Having had a look at some of the assumptions of the interpretations and the choices behind the interpretations, that could often do with more explanation, justification or perhaps reflection, let's turn to the results, discussions and conclusions.
2.3.1	- The results show that the approach proposed by the European Commission to allocate effort sharing is probably the most sensible (focus on capability as ability to pay), because of the variation when other interpretations come in. This is already a worthy finding, and perhaps could be highlighted further. Although, because the overall objective of the paper is perhaps still unclear, it is also unclear whether or not the paper aimed to assess the European Commission's proposed approach for how sensible it is as an approach... I do feel that this is already an interesting, welcome and important finding and I also feel that the authors may not necessarily be happy that this is what they found (am I reading too much between the lines?) since this finding is hardly highlighted. Based on the combination of both a now clear statement of the purpose of the paper in the introduction and newly referring to the embedding within the also historic EU negotiation approach immediately thereafter we now expand much more on a discussion of our findings in that context in the results and conclusions sections. Specifically, we have also more explicitly highlighted the one finding by an additional sentence, which indeed we do think is interesting already on its own. [...] This analysis thus has identified why a capability principle interpretation based on GD/capita ranks so prominently within the EU negotiation process. and in the conclusions section: Our results also can be read as one explanation of why the EU has preferred the capability approach to inform its effort sharing allocation. When EU Member States seek to maximize emission allowances within an equity-compatible range, particularly if only one equity dimension is desired to minimize complexity, the capability dimension emerges as the indicator of choice – across all the potential negotiation space it has by far the highest weight among all dimensions and even when varying across all interpretations. And this is exactly what the EU has done in its 2018 effort sharing regulation and the EC has again implemented in its 2021 proposal.
2.3.2	- I would find the results easier to understand if the variations of potential effort sharing per country were put into the context of the 2030 goal. I.e. what combination of interpretations per member state adds up to the 2030 overarching goal? The authors discuss this somewhat in the text, try to discuss this a bit within the negotiation stances part, but a selection of (graphically-represented) options would be welcome. If the 2030 effort sharing goal is to be achieved, what combinations of interpretations per member state can work? This, of course, if the interpretations are more fully justified/explained as discussed already in the comments above.

	To address this, we have expanded on our introduction of the concept of both the ternary charts and the negotiation points. Contrary to the work by Du Pont and Meinshausen (2018) and Du Pont et al. (2017), now shortly referred to also in the manuscript, we do not test for combinations where each country chooses its own indicator and weighting, but derive negotiation points referring to a uniform choice of interpretations and weighting, applied to all countries, and ensuring that the overall emission reduction target is met (see also our response to comment 3.3.1 below). Our intention is that such results can inform negotiators, on both the implications for their own country, but also on all other countries, when varying these interpretations and weights. Also, there is a question that I feel the authors don't necessarily answer well – can these three interpretations be applied together? The variation in figure 2 is very large and begs this question. If the authors could lay out some scenarios for how these three approaches could/should be implemented together towards the 2030 goal, taking account of various other context considerations, that would be a helpful way of tying several loose ends together. Yes, indeed, this is exactly our intention. We hope to have improved clarity on this issue in the revised manuscript. All corners of the ternary charts represent application of a single dimension, all points along any of the three legs the combination of two of them, with varying weight, and all points within the ternary charts apply all three dimensions, with weights indicated by the respective location. Correspondingly, all negotiation points that are located within the ternary charts do imply an application of all three dimensions, and those located at any of the legs at least two of them. The initial Figure 2 – to show the implication of the individual dimensions and interpretations – does indicate the very broad range that can result, as you indicate. Our concept of the negotiation points is our answer for a concept that does implement the three dimensions together to achieve the 2030 target, in the aggregate, and simultaneously giving the split up by country.
2.3.3	- All the rest of the discussions of the results seems to fall within the realm of 'political processes', 'negotiating stances' etc. But considering the lack of EU political process context given in the paper, the negotiating points that are established via the model do not seem to be embedded within the political reality of the EU system. Under some of the interpretations, some member states could theoretically justify an approach of aiming for no more emissions reductions under the 2030 framework. Given the policy, governance and political processes context, how could this be considered a feasible/fair negotiating stance? Remember the context of policy mixes where bargaining also happens across policy files; the context of aiming for climate neutrality by 2050... Thank you for pointing out this earlier shortcoming. Based on our expansion on the EU effort sharing history (see our response to comment 2.2 above) (as well as our now explicit pointing out the papers objectives, see our response to comment 2.1.4 above) we now close this discussion in the conclusions section by explicitly indicating what the learnings from this approach are, and how negotiators may be informed by these results. We also more critically reflect on both method and results in the conclusions section. In addition to the online tool now available to support, and to just pick a few paragraphs, beyond those further sections already cited above, the conclusions now end with: While we do not analyze the actual political processes, the related governance literature has informed our analysis. First, it identified the particular relevance of path dependency, i.e., the notion that policy decisions once made within a certain frame tend to stick to that

	frame^{10,50}, which guides our choice of past EU effort sharing regulation in one approach to derive negotiation convergence points. Second, the governance literature identifies the relevance of timely transparent information. In particular, the 2019/2021 EU decision of implementing to remain within a Paris compatible carbon budget in combination with the Corona-aftermath and “building back better” may represent a critical juncture in institutional development, at which “decisions of important actors are causally decisive for the selection of one path of institutional development over other possible paths”⁵¹, and conflict over ideas has been identified as important for institutional change⁵². The approach we present is intended to contribute to resolving conflicts over equity perspectives to allow for institutional development. The results of our framework make transparent how different choices of equity interpretations can translate into different country contributions. The range of transparent equity considerations made explicit here allows for appreciating the positions of other countries, as well as for a common understanding of the range of outcomes, and thus can contribute to successful negotiations. For regions without previous effort sharing agreements to refer to, as available for the EU, such transparent and commonly available exploration of the negotiation space is likely to be an even more important ingredient in the process to agree on sharing among subsidiary entities.
2.3.4	- Returning to the point above about the interactions among the interpretations, I would really welcome far more critical engagement with the researchers’ own findings, perhaps by embedding some of their thinking in, and by building on, the literature and politics/policy/governance/process discussions that I have mentioned here. If the authors were to engage critically in thinking on how the various interpretations may interact with each other (e.g. responsibility & renewable expansion with capability), I think the discussion would be far richer. Further critique on the choices the authors make in the construction of their approach and interpretations would also be welcome: what if other considerations were taken into account? The limited critical reflection of the approach presented is also evident from the abstract (lines 16-18), where the authors write that they ‘demonstrate the applicability of our approach’ without referencing critical engagement with the proposed approach. Thanks indeed. We agree, and have revised the manuscript in several dimensions. First, we now embed our approach explicitly within the policy/governance/process discussion (see our response to comment 2.1.1). Second, and based in this discuss the specific contributions within such a process that our approach can deliver and inform about (by having expanded the introduction and the discussion of results). Third, we now extensively discuss also the critical issues, and in particular on the interactions among interpretations (see also our response to comment 2.2.2). For this third area of revision the reflections in the main manuscript – given the word limit restrictions there – are further detailed in the SI (new sections 2-4), now also reflected in mentioning this discussion in the abstract.
2.4	Finally, some small points for correction/clarification:
2.4.1	- It is best to refer consistently to the ‘European Green Deal’ and not the ‘Green Deal’ as there are many different green deals around the world (line 7). Perfect – yes, revised.
2.4.2	- What do the authors mean by referring to the approach as ‘pure’? (line 51)

	Thanks. We have revised this section of the text to now reflect the new structure and content of the revised manuscript, now also avoiding the earlier misleading label.
2.4.3	- Remove time references such as 'current', 'ongoing' (e.g. in Table 1) and replace with specific time phases (e.g. in 2021). Thanks, this clarifies. We revised throughout the manuscript.

References

- Boasson, E. L., & Wettestad, J. (2013). *EU climate policy: Industry, policy innovation and external environment*. Ashgate.
- Capoccia, G. (2016). *Critical Junctures*. In O. Fioretos, T. G. Falletti, & A. Sheingate (Eds.), *The Oxford Handbook of Historical Institutionalism* (pp. 89–106). Oxford University Press.
- Dupont, C. (2016). *Climate Policy Integration into EU Energy Policy: Progress and Prospects*. Routledge.
- Gheuens, J., & Oberthür, S. (2021). *EU Climate and Energy Policy: How Myopic Is It? Politics and Governance*, 9(3), 337–347. <https://doi.org/10.17645/pag.v9i3.4320>
- Gupta, J. (2010). *A history of international climate change policy*. *Wiley Interdisciplinary Reviews: Climate Change*, 1(5), 636–653.
- Hall, P. A., & Taylor, R. C. R. (1996). *Political science and the three new institutionalisms*. *Political Studies*, 44(5), 936–957.
- Jones, E., Kelemen, R. D., & Meunier, S. (2016). *Failing Forward? The Euro Crisis and the Incomplete Nature of European Integration*. *Comparative Political Studies*, 49(7), 1010–1034. <https://doi.org/10.1177/0010414015617966>
- Jordan, A., & Matt, E. (2014). *Designing policies that intentionally stick: Policy feedback in a changing climate*. *Policy Sciences*, 47(3). <https://doi.org/10.1007/s11077-014-9201-x>
- Mavrot, C., Hadorn, S., & Sager, F. (2018). *Mapping the mix: Linking instruments, settings and target groups in the study of policy mixes*. *Research Policy*, 103614. <https://doi.org/10.1016/j.respol.2018.06.012>
- Oberthür, S. (1999). *The Kyoto Protocol. International Climate Policy for the 21st Century*. Springer.
- Peters, B. G., Pierre, J., & King, D. S. (2005). *The politics of path dependency: Political conflict in historical institutionalism*. *Journal of Politics*, 67(4), 1275–1300. <https://doi.org/10.1111/j.1468-2508.2005.00360.x>
- Rhinard, M. (2019). *The Crisisification of Policy-making in the European Union*. *JCMS: Journal of Common Market Studies*, 57(3), 616–633. <https://doi.org/10.1111/jcms.12838>
- Schmidt, V. A. (2010). *Taking ideas and discourse seriously: Explaining change through discursive institutionalism as the fourth 'new institutionalism'*. *European Political Science Review*, 2(1), 1–25. <https://doi.org/10.1017/S175577390999021X>
- Siddi, M. (2021). *Coping With Turbulence: EU Negotiations on the 2030 and 2050 Climate Targets*. *Politics and Governance*, 9(3), 327–336. <https://doi.org/10.17645/pag.v9i3.4267>
- Zaun, N. (2016). *Why EU asylum standards exceed the lowest common denominator: The role of regulatory expertise in EU decision-making*. *Journal of European Public Policy*, 23(1), 136–154. <https://doi.org/10.1080/13501763.2015.1039565>

Reviewer #3 (Remarks to the Author):

I will caveat my comments from the start with: I am from a different discipline and am reading this largely to identify issues regarding logic, clarity, usefulness etc for policy/law/practice. I have gone nowhere near the maths or the methods! If my points are mistaken due to misinterpreting material, then apologies.

The content is interesting, valuable (theoretically and practically) and topical. It builds on three equity principles highlighted by the IPCC and could be very useful. The identification of the 10 components and their application, as well as subsequent analysis provides insights into alternative mechanisms in calculating reduction targets for emissions and possibly identifies foundations for future negotiations (this aspect needs further development/clarification). The piece can contribute to important on-going academic and policy debates and the proposed tool will be very interesting to see also. If the tool ends up being included in the eventual publication, it would be important to ensure that it is more broadly accessible (without having to pay journal fees), i.e. perhaps make it available on an institutional repository or similar (checking with funders and employing institutions first regarding ownership and the like) and include the link in the publication? (In case this is not what is already intended)

Thanks indeed for your careful and comprehensive feedbacks and suggestions, which we respond to one by one below.

We have expanded on how to use this approach as foundation for (future) negotiations, and point out in detail below. On the tool: yes, the tool is available on an open access platform, with the link in the publication. In the revised version this is implemented, and the tool is ready for your access as well at https://kawilliges.shinyapps.io/EU_effort_sharing_tool/ . With publication the tool will be shifted from the personal site to our institutional site, again with open access.

However, improvements could be made in particular regarding clarity and coherency – whether regarding readability, justifications, development of arguments or otherwise.

Overall, I think there is valuable content, but the manuscript needs to be improved and re-worked considerably still. Once the issues are resolved, it has considerable potential to contribute to both academic debate and policy development.

3.1	Clarity: overall, the clarity could be improved across the piece. This relates to both individual sentences and paragraphs, but also the flow of the piece. In particular, if the abstract, introduction and conclusion were improved, this would help the overall piece considerably. However, the same need for clarity applies across the piece, e.g. later in discussing the application of the principles or negotiation convergence points (e.g. lines 278-283). There is also the question to what extent you wish the content to be accessible to individuals from different disciplines. It took a few reads to understand some elements and I am not used to figures and graphs in my discipline, but they were manageable for the main part. Figure 3 could be clearer I think (not my field though) and perhaps continuing to use R1, R2 etc would be useful for consistency? Thank you for pointing out this overall observation. We have substantially revised the manuscript, especially the introduction and conclusions, and expanded on the concept of the negotiation convergence points and its application, seeking to improve accessibility also for general readership. Trying to strike a balance between accessibility and maintaining scientific rigor within the field, beyond consultation between the co-authors we have included other colleagues from neighboring fields by asking for feedback on readability and as such hope to have improved clarity for a broad audience.
-----	--

	Regarding the lines you indicate at the beginning of the section on “negotiation points”: we have changed one of the concepts (see our response to comment 2.1.2) and expanded this introduction significantly to improve clarity. In Figure 3 we have reduced complexity focusing on introducing the concept and restricting the figure to two panels. In this Figure we now include all the shorthand labels for each of the corner interpretations for all panels in the legend, and we also expanded our carrying over R1, R2 etc. more across the paper and in the SI. We have also removed the associated negotiation points, as the Figure comes in the text before these points are discussed, and could potentially confuse readers.
3.2	Purpose: what precisely is the purpose of the paper? Is it theoretical or practical or both? Is it for the EU/Commission/Member States or globally or both/all? What is the end goal? Triggered by your helpful observation we revised the introduction to now explicitly identify the paper's purpose. It is both supplying theoretical results and enabling their use in practical negotiations. The latter is indicated more extensively in the revised conclusions section. In both we also respond to comment 2.1.4 of reviewer#2, and do list the revised text sections in our response to that comment above. We exemplarily apply our method for the EU (informing the negotiations among all the actors indicated, Commission, Council/Member States, and Parliament) but seek to thus show for other regions the applicability for them as well – indicated now better especially in the revised conclusions.
3.3	Role of the principles (related to purpose of paper): how do you see these being used in the end? You mention (briefly/as an aside) some limitations, but still support their use. So who is to use these, how/to what end will/could/ought they apply them (e.g. is there an overarching aim for those applying them), what limitations arise and how can these be addressed... etc? Or does any of this matter? We expanded on the foundation of the principles (section introduction, see also our response to comments 1.2, 2.1.4 and 3.2), made better explicit how the proposed framework can inform negotiators and scientists (in sections results and conclusions, basically all expansions in these sections, shown as underlined in the version of the revised manuscript with track changes), and in particular now discuss the limitations and how to deal with them much more extensively, including also the interaction among interpretations, in the conclusions (basically all the extensions in the conclusions are devoted to that end). Given the word limit, some of this discussion is expanded in more detail in the new sections 2 to 4 in the Supplementary Information, referred to from the main text. See also our response to comment 2.3.4 on this last point.
3.3.1	To build on this: what's to stop individual Member States saying use e.g. R2, C1 and E3 for us please as that involves the least effort/change by them? Would you use different combinations for each Member State? If somehow you apply the 3 principles and you get a perfect balance that is acceptable to everyone (although nobody is happy with a compromise?) and will meet the overall EU targets on the button, what happens if one Member State then doesn't meet their obligations? We have expanded on the objective of our negotiation point analysis to identify weightings within spaces of always the same interpretations across countries (and thus consistent). The approach you indicate has also been covered in the literature (Robiou du

	Pont et al., 2017; Robiou du Pont and Meinshausen, 2018), and we now explicitly refer to it in the manuscript, albeit stating that such diverse interpretation will not achieve the overall goal. We sought to expand the introduction and explanation of our frameworks usability, mainly in the results and conclusions sections, but also by the new embedding given in the introduction. On the last point, i.e. if countries later do not fulfill their commitment, we would consider this a monitoring and sanction issue, which would go beyond the specific objective of our framework, and equally apply to other frameworks that could be applied.
3.3.2	The paper doesn't need to address everything, but it needs to be clear as to what it is seeking to address, the limits it is imposing on itself (write a follow-up paper if needed!) and the caveats that need to be taken into consideration. We have now explicitly addressed the objectives of this paper (thanks for and see also our response to comment 3.2 above). In particular we have expanded strongly on and now much more explicitly discuss the limits and caveats, in particular in the many new paragraphs in the conclusions, partly also referring to details in the SI (sections 2-4). Thanks, this constitutes a very crucial improvement.
3.4	Justifications: these are frequently lacking and would be of assistance. Including: why this paper, why this focus, why these principles, why these facets and not others... this applies across the paper. Thanks for this reflection. We accordingly have considered it in revising our proposed frameworks embedding (the introduction and new effort sharing section). In the introduction and in the section "Equity principles" we now have expanded our reference to the base of the selection of principles, the IPCC, by additional sentences in each of them. Equally importantly, we now explicitly discuss further interpretations for these principles, and which of them we have included for what reasons (with further details in sections 2-4 in the SI). To that end we have expanded by two more interpretations in the main text (and discuss further ones in the SI), yet still clarifying the pros and cons for them. Of those changes, to just pick the revised embedding, the introduction now includes: In political processes, quite often this effort sharing of a common target is hampered by diverging viewpoints and perspectives on how to determine a fair share. Building upon the categorization of equity principles used in the effort sharing literature⁶⁻⁹ we present an approach that can contribute to a transparent decision process when partitioning an overall emissions target among subsidiary entities by aligning disparate views and defining an allocation space where different parties could agree to an equitable compromise. For negotiation processes heavily influenced by a range of factors, such as historical path dependencies¹⁰, side deals, power struggles, different cultures, or domestic politics, such transparency in terms of equity may be a significant factor contributing to success, given the increasing weight of the equity dimension in such negotiations¹¹. We assess the implications of such an approach – in terms of the resulting emissions budget allocation and corresponding reduction targets – in the context of the European Union (EU) negotiations in 2022 to raise the 2030 effort sharing target¹². While in 2021 an overall target of 55% reduction compared to 1990 was established, distribution of that effort among Member States is undecided and under negotiation in 2022. We begin by exploring the implications of different effort sharing mechanisms and interpretations of equity or justice on a singular, country level, finding that the resulting ranges of possible

	emissions reduction burden can vary widely. In such a case, if each country were to favor a different equity interpretation, the overall target of the EU in 2030 would most likely remain unmet^{13,14}. However, our proposed framework allows for systematically combining different interpretations of equity or justice on the basis of three major principles – capability, equality, and responsibility – and allows us to explore situations where country targets are the result of a weighted combination of interpretations of those principles, where member state reductions add up to the overall EU reduction target. In doing so we distinguish interpretations of equity or justice that reflect considerations of equality, capability and responsibility. By identifying weighted combinations of interpretations of these principles that minimize the changes in emission reduction effort required by countries compared to their (i) upper bounds of equity-compatible emissions or, given the practical importance of negotiation history, (ii) previous commitments in the earlier less ambitious effort sharing agreement, we identify a possible space for decision-making which combines multiple equity interpretations. We find that the possible combinations of equity interpretations, and the strength at which they are applied, result in a wide range of space for decision making which comprises a richer set of ethical considerations than the 2021 proposed EU approach. We conclude with a discussion of the described framework and its applicability in specific negotiation and policymaking processes, also in other international contexts.
3.5	Proofread: There are some slips/typos across the piece and a careful proofread will be required at the end. E.g. Pg2, line 22, ‘need’ not ‘needs’. ‘Broken down to’? ‘Broken up and redistributed across’ perhaps? The slips also affect the clarity at times. Thank you. In a careful proofread we sought to have taken care of all of them.
3.6	Some more specific points:
3.6.1	Why is C1 meant to mirror the EU approach? Given that the EU negotiation history (now also covered/reflected in a new section following the introduction) relied so strongly on the capability criterion, we sought to have one interpretation of this dimension in our analysis as well that closest mirrors this EU approach. We do so by a statistical approach set forward in methods to arrive at C1. The motivation for this reliance on history and the earlier agreements within the EU is now given in much more detail at various places, e.g. also at the above mentioned introduction of the negotiation points (at the beginning of the such labeled section). This we hope makes it much easier accessible also for a general readership.
3.6.2	Are there other facets for each of the three principles? Why discount these? E.g. could you just say for equality: each State should reduce their outputs by X%? Or by size of the country? Similarly, for responsibility, why not go back longer and look at the harms done (rather than benefits garnered) irrespective of the timing of scientific reports? We have expanded on further facets, to make the discussion more complete. In more detail: An emission reduction at an equal level across countries is our starting point, that we then modify by any of the other considerations. In Figure 4 we now do mark this “fall back” position of equal reduction targets across countries for the effort sharing sector (dotted line, explained in the figure caption). By the way, this would not be fair, as it takes the starting level of emissions for granted (as we argue in more detail in (Williges et al., 2022)). Given that the emission reductions are specified in rates relative to 2005 emissions, absolute emission reduction perfectly vary with country size (if size is measured by emission level). While on the capability dimension we have not varied by

	other indicators of country size, we did add two more indicators for this dimension (renewables also for capability; governance) - these now specified in Table 1, the according text introduction, and results covered in Figures 2-4, and respective results discussion. On the time horizon for looking backward: yes, indeed, this is ambiguous. We thus have integrated a sensitivity analysis for switching the year up until we look back to, varying from 1990 to 2000. Results indicate, that only for very few countries (7) there is a change, and only for a fraction of them one that reaches the order of magnitude of 10% deviation. We conclude that the main issue is rather whether to consider historic responsibility than the particular year chosen for actual implementation. (SI section 3, and discussion of these results in the conclusions section). To cite just on the last issue from the revised manuscript: First, when introducing the year of reference in the “equity principles” section: One could cite other dates for good reasons, e.g. the 1992 ratification of the UNFCCC at the Earth Summit or the late 1980’s establishment of the IPCC, which we assess in more detail in the Conclusions section and in the Supplementary Information, section 3. Second, a discussion in the conclusions section: Beyond choice of an interpretation, its application can have varying effects on outcomes. The most obvious example is the question of when to start taking historical emissions into account; we choose 1995 based on publication of the IPCC’s Second Assessment Report. However, valid arguments can be made that other, particular earlier, years should be chosen. We find that while that is the case, shifting the year has less of an effect on outcomes compared to the initial choice to consider countries’ historical responsibility. Overall emissions reductions for countries are mostly unaffected or would see only minimal changes (see Supplementary Information, section 3 and Figure SI.2 for further detail). For the specific case of the EU, both the historical emissions and inherited benefits (R1 or R2) interpretations acknowledge the specific context at the time, with Eastern European countries having been comparatively emission intensive at lower efficiency up to 1990 with emissions plummeting thereafter. Historic emissions consider aggregate emissions over the whole period back to 1995, but not including the high-emissions period up to 1990. Benefits received are derived from capital stock available in 1995 (i.e., after economic restructuring), and are evaluated using a recent average EU emission intensity (and not the historical – and more emission intensive – levels from the years before 1990). as well as have included the more detailed discussion and respective figure in the Supplementary Information (section 3).
3.6.3	How different is E2/E3 from R3? The E2/E3 interpretations are forms of per capita convergence; beginning with unequal emissions levels, they describe pathways where emissions per capita converge across countries to an equal per capita rate by 2030. Thus, countries with today higher per capita emissions are allocated a large share of the remaining EU carbon budget. R3, on the other hand, implements an equal-per-capita approach from the very beginning, and maintains the same distribution until 2030. In other words, under R3 the total EU carbon budget remaining up to 2030 is split to give an equal share per capita to all European inhabitants. Thus, it does not introduce any form of grandfathering, of which the same cannot be said for E2 and E3.

	The entries for R3 and E2/E3 have been clarified to reflect that R3 does not include grandfathering, while E2/E3 begin at unequal levels and converge to an equal rate.
3.6.4	Are there not concerns that some approaches could hinder overall efforts? Eg. R4 rewards past good behaviour by allowing increased levels of promotion? (I know this is reflected in several approaches, but it's more blatant here). Yes, indeed. This interesting discussion is now explicitly taken up in the conclusions section (and with some reference to sections 2-4 of the SI). We note – now also in the manuscript – that incentives and levels of hindering also differ with respect to the interpretation one and the same indicator can be used for (we discuss for renewable extension in the manuscript – elaborating on your example, as indeed, it may be the most blatant one.) For example, the new discussion in the conclusions section includes: We note that the results of our approach can be employed to inform parties in ongoing political processes but do not serve as a projected end-point for negotiations, due to a number of factors. Our aim is to provide a framework for distributing future emissions budgets based on well-established equity principles, but it must be emphasized that contextual factors, the choice of interpretations and their implementation can lead to differing outcomes. Numerous potential proxies could be suggested for a given interpretation (see Supplementary Information section 2 for discussion on possible governance indicators). Application of increasing renewable shares, as either a responsibility or capability interpretation, is an apt example; valid arguments can be made to place it under either equity consideration, but arguments against are also relevant. Implementation of an RES-based capability interpretation is problematic as it can be seen to reward countries for a lack of past effort without encouraging lagging countries to act by increasing their reductions. Also, past performance regarding RES is not a reliable indicator of future performance, as political or economic conditions may lead to changing capability. For a further discussion on the ambiguity of renewables as an equity interpretation, see the Supplementary Information, section 4.
3.6.5	Re basic needs –why multidimensional poverty? Why not simply needs or heating needs? (not objecting to this, but wanting the justifications) Thanks for pointing out. We have expanded and included an additional sentence in Table 1 explaining our use of a multidimensional poverty index. Please see also our response to comment 2.2.4 of reviewer#2 above.
3.6.6	The different number of components for each principle is interesting – e.g. if seeking to weight R, E and C equally and using all components, does that lessen the significance of R1-5 for instance as there are 5 compared to 2 C and 3 E. We have expanded our introduction of the dimensions concept (the three principles, with this term used in the manuscript consistently) based on various interpretations each (and yes, with different numbers so, now 3-5 for each). This refers to the section “Combinations of equity interpretations” and beyond. We hope to have made it better explicit that our approach always combines each of the three principles (equality, capability, responsibility), yet with possibly a weight of zero for up to two of them, but in any case for each of these principles has to choose one specific interpretation (and thus indicator) only. We hope that in the revised section on “negotiation points” this is now better evident, that we are identifying combinations of weights for these three principles,

	for each and every combination of interpretations possible (but always only one interpretation for each principle).
3.6.7	What is a cathete? After some googling and consulting with some mathematicians, I think you mean cathetus (cathète in French)? But the mathematicians I spoke with don't use the term. Thanks, we revised to the correct terms in English (base and legs for equilateral triangles).
3.6.8	Does the following point just hold for Germany or for every country? Either way, why? 'While increasing the weight of equality (i.e., moving down along the right hand cathete) reduces emission reduction targets – the reason being that an increase in the weighting of this dimension necessarily decreases the weight of the other two dimensions, each of which imply stronger reduction target increases.' We added in the exemplary presentation of the ternary chart for Germany that this is an empirical finding just for Germany (manuscript, just before Figure 3). [...] each of which we find empirically for Germany to imply stronger reduction target increases. The reason is that for these specific interpretations of R, C, and E the reduction target for Germany increases when raising the weight of R and/or C. When increasing the weight of E, weight has to be reduced for at least one of the others, and as both of these other two dimensions mean stronger reduction (for Germany) than the equity dimensions does, the overall reduction target for Germany has to decline when moving that direction. We also added in the manuscript: Shifting the weights among the three dimensions or switching the interpretation used for each corner has a different influence on emission reduction targets depending on the country and interpretations assessed; a set of ternary charts illustrating the effects of changing all interpretations for all countries can be found in the Supplementary Material, section 5, or can be generated via use of an interactive web tool developed for this framework (for details, see https://kawilliges.shinyapps.io/EU_effort_sharing_tool/).
3.6.9	It was meant to be possible to combine multiple or all factors – was this done? Is it useful? Rather than having numerous triangles for each country using just 3 components? Our framework allows for combinations of the equity considerations by weighting one interpretation from each corner (Responsibility, Capability and Equality). We have substantially revised our Results sections “Implementing combinations of equity interpretations” and “Possible negotiation convergence points” to better explain how our process works and its usefulness, while focusing on the limits and where we see its novelty and applicability. We do not combine interpretations of one principle (i.e. there is no synthesis of all 5 responsibility interpretations etc.) as our aim was to demonstrate the approach and a range of possible interpretations without being exhaustive in presenting all possibilities.
3.6.10	What about using all R components and nothing from E or C for instance?

	We do not investigate such a scenario, as we focus on developing allocations based on combining the three equity principles (R, E and C) as laid out in IPCC AR5 WGIII Ch. 6 (Clarke et al., 2014) (and discussed at length in chapters 3 and 4). However, combinations of R, E, and C where only one component is weighted 100% is assessed in our work, and as can be seen in Figure 4 it does occur that a 100% weighting for a capability interpretation could be a possible negotiation point. So, in reference to the answer to your previous comment above (3.6.9), it would be possible that an R component that is itself a combination of the R interpretations we present could result in a 100% weighting as a negotiation point (but note this is a purely hypothetical example, and not likely given our results' tendency to favor the other two equity principles). We do not investigate combinations of interpretations that reflect only one equity principle for the reasons discussed in response to comment 3.6.9, but leave it to possible future analysis whether and if how combinations of interpretations of only one principle can be applied, as this would raise a different set of normative issues, namely the issue of which interpretations of one and the same principle meaningfully can be combined.
3.6.11	The graphs supposedly enable the identification of country preferences – what do you mean (existing preferences or it allows countries to figure out what suits them best based on either criteria or the outcome?) and how does it do this? The explanation was fully rewritten, and should be much easier accessible now. The intention was to indicate that countries are likely to seek for as small of a difference to already planned policy, and thus to past negotiation results. This is now explained in the section “negotiation points”, also using improved labels for the two concepts (see also our answer to comment 2.1.2 of reviewer #2 above).
3.6.12	How does this facilitate ‘identification of possible points of agreement’? (pg13, c. line 256) This is further developed on pg16, but some greater clarity is needed and cross-referencing from earlier on to the later section would be useful. We rewrote the respective section on introducing this concept, now describing in more detail (section on “negotiation points”). More importantly, we deleted this pre-reference at the location you mention in this comment (i.e. we deleted to make reference to it before the section that actually introduces the concept), as it was both easily confusing and not really necessary here already.
3.6.13	What do you mean by ‘these points indicate the weighted combination of equity interpretations which result in the lowest possible aggregate deviation of all EU countries from a given preference, e.g. the equity consideration which requires minimal mitigation effort.’? What is the ‘given preference’ here? Are you presuming that each Member State wants to make the least effort possible? How are you balancing this out between Member States? E.g. if countries with high emission levels (in absolute terms, not per capita) make very little effort, then other countries will have to make a disproportionately large effort. This comment addresses a similar issue as the one addressed by reviewer #2, comment 2.1.2, and we agree with both of you, we did not adequately frame our optimization and mislabeled it as ‘minimal effort.’ We have significantly changed the text to reflect that what we aim to do is minimize the change in country emissions compared to either the

	2018 Effort Sharing Regulation or to a second yardstick, the upper bound of equity-compatible emissions. (see also the more extended answer above on 2.1.2)
3.6.14	I'm not sure how the minimal effort is being identified for one route, but for the one linked to the ESD, the results are not simply unsurprising – they are surely expected? If one calculates the minimal effort relative to the ESD and that is largely based on capabilities, the minimal effort is clearly also going to be largely based on capabilities. It's not exactly circular, but close enough. Regarding the other route: if this is linked to existing practices/measures, presumably many EU countries nowadays do what they feel they are able to? And EU policy doesn't appear overnight – MS would have been preparing and also influencing its development, as well as in light of international commitments more generally... Minimal effort seems to be something that is likely to be linked to capability... Should minimal effort be your focus? We agree that policies do not appear quickly, and member states have been preparing, particularly with regard to the 2018 ESR; it seems reasonable then to assume that a starting point for agreement on a new effort-sharing allocation would be one which minimizes the overall changes compared to the old agreement, in order to limit the need for new and unexpected policy. In order to clarify our intent, we have changed the results and conclusions sections and revised to eliminate minimal effort and instead emphasize issues of path dependency. These changes occur throughout the sections. Regarding the specific sentence you highlight in your comment (occurring just before Figure 3), we have re-written the section to improve clarity, as follows: Calculating the resulting emissions reductions requirement of any given combination of three interpretations for all EU countries provides a wealth of information for countries in terms of their negotiation position – considering the maximum emission points in the charts represent the upper bound of equity-compatible emissions – but also the rate of change in reduction levels as the weighted combination moves away from such a point or isoline. Combining this information on all EU Member States makes possible the identification of possible points of agreement in future effort-sharing negotiations, discussed in further depth in the next section. We revised and expanded the introduction of the negotiation point concept: In negotiating an effort sharing agreement, agents (in our case EU Member States) could be motivated by a number of aims. One could be to minimize deviations from planned reductions as a result of established policies. Implementing such a target in our analysis would mean that when determining national budgets to 2030 using a single interpretation of each of the three equity principles from Table 1, a weighted combination of the three can be identified that ensures that countries have to do the least additional effort beyond what they agreed in 2018 for their respective reduction by 2030. They may want to keep planned reductions as close to this prior agreement as possible, in order to avoid sudden drastic changes in requirements or policy. In this case, a combination of interpretations can be identified which minimizes the aggregate effort of all EU countries beyond their prior agreement. Formally, the sum of squares of these deviations is minimized (see Methods). We define each of these weighting combinations “negotiation points” and calculate one for each possible combination of the three equity criteria interpretations discussed (60 in all). As a second metric for comparison, we also minimize aggregate deviation from what is the upper bound of equity-compatible emissions for each country,

	i.e. an emission level that can be considered equitable by at least one interpretation. The results of these calculations for both cases, namely, a ‘least deviation from past share allocation’ (blue points, corresponding to minimizing the deviation compared to the 2018 ESR) and a ‘least deviation from the upper bound of equity-compatible emissions’ (marked in orange), are identified in the ternary inset panel in Figure 4. Note that results of the latter are robust against the integration of any further equity interpretations as long as our interpretations cover the overall possible range, a goal which guided their selection (see SI, sections 2-4). Each point in the inset panel represents a combination of one each of an equality, responsibility, and capability interpretation (applied to all countries) which meets the 55% EU reduction target. The ternary subpanel of Figure 4 shows that these negotiation convergence points span the negotiation space, indicating a variety of combination weightings which could likely result from negotiations if Member States follow this rationale. Some trends do emerge. [...] In addition, to emphasize that EU policy does not appear overnight, we introduced a new section on the history of EU effort sharing (the very first section after the introduction) and revised and expanded the conclusions section to discuss how the approach here suggested can be used in the negotiation process (on this, compare also our responses to reviewer#1, comment 1.1, and reviewer#2, comment 2.1.1).
3.6.15	Is the logic: countries will want to make the minimal efforts possible; their minimal efforts that might still facilitate overall EU aims (this argument is weak, as seems to be based on existing balance and therefore quasi-pre-determined/biased – both re quantum and approach) can be identified using the 3 principles; specifically, those minimal efforts are typically based on X, Y and Z (e.g.C1); this common approach/understanding facilitates negotiations and agreements.....? We hope we have addressed this in the previous comment with our discussion on ‘minimal effort’ and have addressed it in the manuscript by clarifying our optimization approach (and removing the concept of minimal effort) in the Results, Conclusions, and Methods sections of the text.
3.6.16	Fairness is mentioned on p.17- is that the overall goal? How does it work with the principles? Thanks. We now made the overall goal better explicit, both in the introduction (see also responses to reviewer#1, point 1.1 and reviewer#2, point 2.1.4) and on fairness in particular in the section “equity principles”. In the latter to explicate how it works with the principles, we expanded In analyzing how the EU emissions budget 2020-2030 can be allocated among the EU 27 we distinguish three principles, namely – following the IPCC’s broad classification¹⁸ – capability, equality, and responsibility and different interpretations of each of these principles. by the following: According to this understanding, these principles can be interpreted differently as they can reflect different ethical considerations. Different interpretations of these principles and combinations of them amount to different interpretations of what allocation can be

	considered equitable or just. Our approach builds upon the state of the art – in particular as reflected in the IPCC, which is focused on informing policy makers – and aims to assess the implications of imposing equity or justice principles in negotiations on the allocation of emission reductions within the EU.
--	--

References:

- Clarke, L., Jiang, K., Akimoto, K., Babiker, M., Blanford, G., Fisher-Vanden, K., Hourcade, J.-C., Krey, V., Kriegler, E., Löschel, A., McCollum, D., Paltsev, S., Rose, S., Shukla, P.R., Tavoni, M., van der Zwaan, B., van Vuuren, D.P., 2014. Assessing transformation pathways, in: Edenhofer, O., Pichs-Madruga, R., Sokona, Y., Farahani, E., Kadner, S., Seyboth, K., Adler, A., Baum, I., Brunner, S., Eickemeier, P., Kriemann, B., Savolainen, J., Schlömer, S., von Stechow, C., Zwickel, T., Minx, J.C. (Eds.), Contribution of Working Group III to the Fifth Assessment Report of the Intergovernmental Panel on Climate Change. Cambridge University Press, Cambridge.
- Kolstad, C., Urama, K., Broome, J., Bruvoll, A., Cariño Olvera, M., Fullerton, D., Gollier, C., Hanemann, W.M., Hassan, R., Jotzo, F., Khan, M.R., Meyer, L., Mundaca, L., 2014. Social, Economic and Ethical Concepts and Methods, in: Edenhofer, O., Pichs-Madruga, R., Sokona, Y., Farahani, E., Kadner, S., Seyboth, K., Adler, A., Baum, I., Brunner, S., Eickemeier, P., Kriemann, B., Savolainen, J., Schlömer, S., von Stechow, C., Zwickel, T., Minx, J.C. (Eds.), Climate Change 2014: Mitigation of Climate Change. Contribution of Working Group III to the Fifth Assessment Report of the Intergovernmental Panel on Climate Change. Cambridge University Press, Cambridge, United Kingdom and New York, NY, USA.
- Robiou du Pont, Y., Jeffery, M.L., Gütschow, J., Rogelj, J., Christoff, P., Meinshausen, M., 2017. Equitable mitigation to achieve the Paris Agreement goals. *Nat. Clim. Change* 7, 38–43. <https://doi.org/10.1038/nclimate3186>
- Robiou du Pont, Y., Meinshausen, M., 2018. Warming assessment of the bottom-up Paris Agreement emissions pledges. *Nat. Commun.* 9, 4810. <https://doi.org/10.1038/s41467-018-07223-9>
- Williges, K., Meyer, L.H., Steininger, K.W., Kirchengast, G., 2022. Fairness critically conditions the carbon budget allocation across countries. *Glob. Environ. Change* 74. <https://doi.org/10.1016/j.gloenvcha.2022.102481>

Reviewer comments, second round of review

Reviewer #1 (Remarks to the Author):

The authors have addressed all my previous comments satisfactorily, I have nothing further to add.

Reviewer #2 (Remarks to the Author):

Review of revised article

'Sharing the effort of the European Green Deal among countries'

Claire Dupont

Well done to the authors on dealing with the extensive comments and feedback of the reviewers in such a considered manner. I do feel that the paper reads much better: the aim is clearer and the scope and relevance of the approach is presented in a far more nuanced way. The updates and additions made to interpretations of the principles go a long way to alleviate my concerns about the paper.

Furthermore, I really appreciate the expanded conclusions. The reflection of the authors on the relevance of their paper, on the implications of the choices they made in their approach, in highlighting the limits of the approach, while also highlighting the potential for application in the EU and in other contexts is well done. The paper seems far more valuable a contribution with these reflections added.

I recommend the paper be published, after the authors make a final check to correct the tiny omissions or any slight typing mistakes.

Here, I list a few small points that I spotted, which I hope helps in the final edit phase. (Based on the track changes version):

- Perhaps add a sentence or a few words in the abstract connecting to the broader international context now mentioned in the conclusions rather than focusing relevance of the paper only on the EU 2030 context.
- p. 3 'While in 2021 an overall target of 55% reduction by 2030' (add 'by 2030')
- p. 7 bottom of page should read 'ability of an actor to effect changes' (not ability 'for')
- p. 15, you write 'less emissions reduction', I wonder would it not be more correct to write 'lower levels of emissions reduction'...? Also when you write 'less emission reduction burden', I think it would be better to write 'lower emission reduction burden'.
- P. 16, the phrasing 'stricter reduction targets' could also be written as 'higher reduction targets' potentially.
- P. 27 'the notable two exceptions...' would usually be written 'the two notable exceptions'.
- P. 34 under methods: '4four' – to be corrected.

Responses to reviewer comments on revised submission of article titled:

Sharing the effort of the European Green Deal among countries – *Nature Communications*

Dear Claire Dupont, thank for your considerations; we are honored by your evaluation. The very detailed and considerate reviewer comments, and in particular yours, have indeed been the basis for this – we also see it as significant – improvement.

Thanks for the remaining issues you spotted. We list below how we implemented them (again in the table below reviewer comments are *in italics*, and responses are **blue colored text**, with direct citations from the revised manuscript **in green colored text**).

In addition, we supply a file of the manuscript where all changes to the first revision of the manuscript are marked (track change). This file, however, does in addition also indicate the changes in response to the suggestions by the editorial-guided author check-list (such as a restructuring of the order in the introductory section and deletion of subheadings there).

Sincerely,

The Authors

REVIEWER COMMENTS

Reviewer #2 (Remarks to the Author):

Review of revised article

'Sharing the effort of the European Green Deal among countries'

Well done to the authors on dealing with the extensive comments and feedback of the reviewers in such a considered manner. I do feel that the paper reads much better: the aim is clearer and the scope and relevance of the approach is presented in a far more nuanced way. The updates and additions made to interpretations of the principles go a long way to alleviate my concerns about the paper.

Furthermore, I really appreciate the expanded conclusions. The reflection of the authors on the relevance of their paper, on the implications of the choices they made in their approach, in highlighting the limits of the approach, while also highlighting the potential for application in the EU and in other contexts is well done. The paper seems far more valuable a contribution with these reflections added.

I recommend the paper be published, after the authors make a final check to correct the tiny omissions or any slight typing mistakes.

2.1	Reviewer #2 (Remarks to the Author): Here, I list a few small points that I spotted, which I hope helps in the final edit phase. (Based on the track changes version): - Perhaps add a sentence or a few words in the abstract connecting to the broader international context now mentioned in the conclusions rather than focusing relevance of the paper only on the EU 2030 context.
-----	---

	Thanks. Yes, we revised the abstract to include this [for many potential readers] crucial information. The abstract now ends with the following (and is – to follow the overall length guideline – correspondingly slightly shortened before): Whereas we apply our approach within the setting of the EU negotiations, the framework can easily be adapted to inform debates worldwide on sharing mitigation effort among subsidiary entities.
1.2	- p. 3 'While in 2021 an overall target of 55% reduction by 2030' (add 'by 2030') We added. - p. 7 bottom of page should read 'ability of an actor to effect changes' (not ability 'for') We changed. - p. 15, you write 'less emissions reduction', I wonder would it not be more correct to write 'lower levels of emissions reduction'...? Also when you write 'less emission reduction burden', I think it would be better to write 'lower emission reduction burden'. We did change following both suggestions. - P. 16, the phrasing 'stricter reduction targets' could also be written as 'higher reduction targets' potentially. While we followed all other suggestions, here we were too concerned that in some places "higher emission reduction" might be confused with "higher emission level", and thus might be potentially misleading. We thus preferred to remain with "stricter". - P. 27 'the notable two exceptions...' would usually be written 'the two notable exceptions'. We corrected. - P. 34 under methods: '4four' – to be corrected. This was an artefact of the "track changes mode" only – the deletion of the number "4" was not easily visible (horizontal deletion line).